# DOS: DREAMING OUTLIER SEMANTICS FOR OUT-OF-DISTRIBUTION DETECTION

## ABSTRACT

Detecting out-of-distribution (OOD) samples is essential when deploying machine learning models in open-world scenarios. Zero-shot OOD detection, requiring no training on in-distribution (ID) data, has been possible with the advent of vision-language models like CLIP. This scenario presents a more practical alternative than traditional OOD detection. By building a text-based classifier with only closed-set labels, the model can achieve impressive OOD detection performance. However, this largely restricts the inherent capability of CLIP to recognize samples from large, open label space, making it insufficient to detect hard OOD samples effectively. In this paper, we provide a new perspective to tackle the constraints posed by exclusively employing closed-set ID labels in zero-shot OOD detection. We propose leveraging the expert knowledge and reasoning capability of large language models (LLM) to Dream potential Outlier Semantics, termed DOS, without access to any actual OOD data. Owing to better consideration of open-world scenarios, DOS can be generalized to different OOD detection tasks, including far, near, and fine-grained OOD detection. Technically, we design (1) LLM prompts based on visual similarity to generate potential outlier class labels specialized for OOD detection, as well as (2) a new score function based on the proportionality between potential outlier and ID class labels to distinguish hard OOD samples effectively. Empirically, our method achieves new state-of-the-art performance across different OOD tasks and can be effectively scaled to the large-scale ImageNet-1K dataset.

## 1 INTRODUCTION

Machine learning models excel in closed-set scenarios, where training and testing datasets share identical distribution. However, in open-world settings, especially in high-stakes scenarios like autonomous driving where the consequence of making an error can be fatal, these models often encounter out-of-distribution (OOD) samples that fall outside the label space of the training dataset, leading to unpredictable and frequently erroneous model behaviors. Consequently, there is a growing interest in OOD detection (Yang et al., 2021; 2022; Salehi et al., 2021), aiming to distinguish OOD samples from test-time data while maintaining classification accuracy.

Most existing OOD detection methods (Hendrycks & Gimpel, 2017; Lee et al., 2018; Hendrycks et al., 2019b; Liu et al., 2020; Sehwag et al., 2021) can effectively detect OOD samples based on a well-trained in-distribution (ID) classifier. However, they are constrained to ID datasets with different label spaces. Besides, these methods solely depend on vision patterns, ignoring the connection between visual images and textual labels. Recently, Ming et al. (2022) introduced the setting of zero-shot OOD detection, which aims to leverage the capabilities of large-scale vision-language models (VLMs), *e.g.*, CLIP (Radford et al., 2021), to detect OOD samples across diverse ID datasets without training samples. By constructing a textual classifier with only ID class labels, Ming et al. (2022) achieves impressive performance compared to traditional OOD detection methods.

However, such an approach often fails when encountering hard OOD samples, as shown in Figure. 1 (a). One might wonder 1) if this issue arises because the pre-trained models (*e.g.*, CLIP) are not strong enough or require further fine-tuning; or 2) if it is attributable to the usages of these pre-trained models, *e.g.*, *an exclusive reliance on closed-set ID classes*. Surprisingly, our findings suggest that CLIP can achieve superior OOD detection results (as depicted in Figure. 1 (b)) by incorporating actual OOD class labels. This reinforces that relying solely on ID class labels is inadequate for

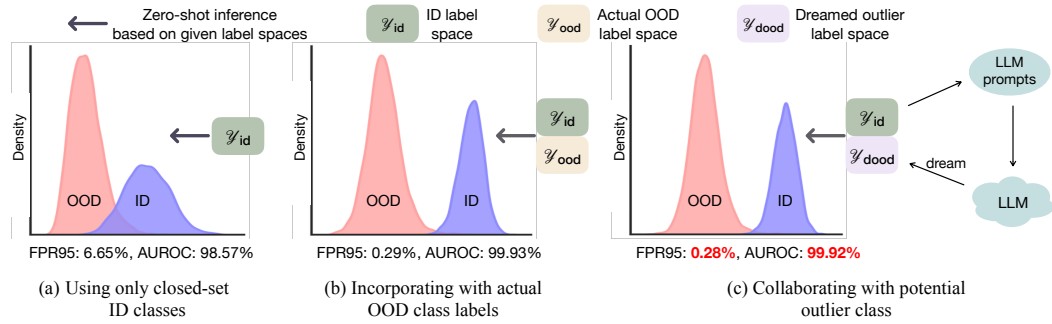

Figure 1: Comparison of zero-shot OOD detection score distribution. Compared to the model using only closed-set ID classes (a), adding actual OOD class labels (b) can largely increase the OOD detection performance. By adding the outlier classes generated by our method (c), the OOD detection results can also be significantly improved without using the actual OOD class labels. We use CUB-200-2011 (Wah et al., 2011) as ID classes and Places (Zhou et al., 2017) as OOD classes.

distinguishing hard OOD samples. Unfortunately, we are unable to access the actual OOD label space in practical open-world scenarios. Therefore, we raise an open question:

*Is it possible to generate the potential outlier class labels for OOD detection without access to test-time data?*

To answer this question, we take a step further in this work and ponder whether we can employ large language models (LLMs) to address this challenge. We propose a knowledge-enhanced approach that harnesses the expert knowledge and reasoning capabilities of LLMs to Dream potential Outlier Semantics, termed DOS, without relying on any actual or auxiliary OOD data, as shown in Figure. 1 (c). Technically, we design LLM prompts to generate potential outlier class labels specialized for OOD detection, following a visual similarity rule. For example, "*Give three categories that visually resemble a horse*", in which horse is an ID class. Furthermore, we introduce a new scoring function based on the proportionality between potential outlier and ID class labels to distinguish hard OOD samples effectively. Significantly different from ZOC (Esmaeilpour et al., 2022) and CLIPN (Wang et al., 2023) that also attempt to generate "NOT ID" classes, our DOS neither requires additional training on a text-based image description generator (as ZOC) nor necessitates an extra dataset to train the CLIP architecture (as CLIPN).

The proposed DOS brings significant performance improvements and enjoys the advantages of: (1) **OOD-Agnostic**, which does not require any prior knowledge of the unknown OOD data; (2) **Zero-Shot**, which serves various task-specific ID datasets with a single pre-trained model; (3) **Scalability and Generalizability**, which effectively scales to large-scale datasets such as ImageNet-1K (Deng et al., 2009) that it is flexible and generalizable across far, near, and fine-grained OOD detection tasks.

Our contributions can be summarized as follows:

- We propose a new paradigm, called DOS, which leverages expert knowledge from LLM to dream potential outlier class labels for zero-shot OOD detection.
- We provide three LLM prompts to dream potential outlier class labels for OOD detection, which are applicable to a variety of datasets within far, near, and fine-grained OOD detection tasks.
- We design a new score function based on the proportionality between potential outlier class labels and ID class labels, helping the model effectively distinguish between ID samples and OOD samples.
- Our DOS is superior, significantly outperforming the strong baseline. Without an increase in inference time, DOS achieves improvements of 2.47%, 1.80%, 7.12%, and 12.77% on the far OOD, near OOD, fine-grained OOD, and ImageNet-1K far OOD detection tasks in terms of FPR95.

## 2 PRELIMINARIES

**Contrastive Language-Image Pre-training (CLIP)** (Radford et al., 2021) is trained on 400 million (image, text) pairs collected from the internet using self-supervised contrastive representation learn-

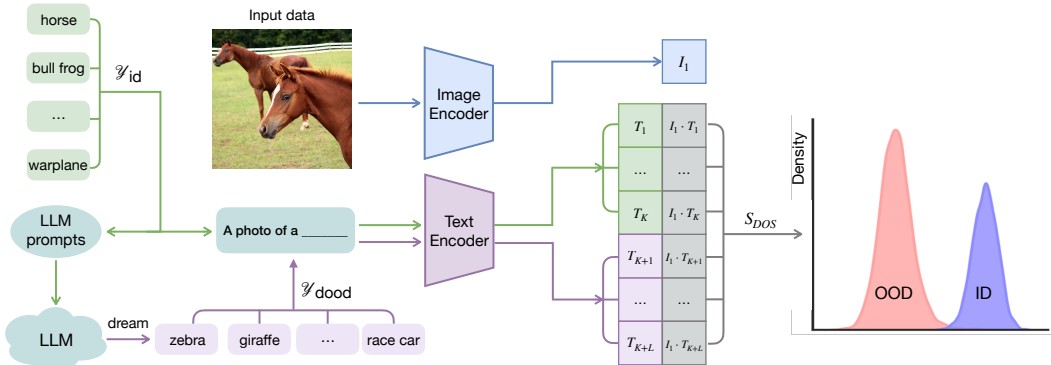

Figure 2: The framework of the proposed DOS. Given a set of ID class labels $\mathcal{Y}_{\text{id}}$, we first leverage the designed prompts to generate a set of outlier class labels, $\mathcal{Y}_{\text{dood}}$, by using a LLM. Then, we input both the ID and generated OOD class labels into the text encoder for building the textual classifier. During the test stage, given an input image, we obtain the visual feature by the image encoder and calculate the similarities between the visual feature and the textual classifier. Finally, the OOD score is obtained by scaling the similarities with the proposed detector score function $S_{\text{DOS}}$.

ing (Tian et al., 2020). The model comprises an image encoder $\mathcal{I}(\cdot)$, adopting either ViT (Dosovitskiy et al., 2021) or ResNet (He et al., 2016) architecture, and a text encoder $\mathcal{T}(\cdot)$, utilizing the Transformer (Vaswani et al., 2017) architecture. During testing, the model evaluates the similarity of visual and textual features to choose the best-matching class label. CLIP enables open-set textual inputs as class labels without retraining or fine-tuning based on specific training data, making it widely applied to zero-shot downstream tasks, such as visual classification, detection, segmentation, and so on.

**Large Language Models (LLMs)** refer to natural language processing models trained on massive data, with more than hundreds of billions or more parameters, such as GPT-3 (Brown et al., 2020) GPT-4 (OpenAI, 2023), PaLM (Chowdhery et al., 2022) and LLaMA (Touvron et al., 2023). These models demonstrate the proficiency to understand and generate natural language text, thus enabling them to undertake a multitude of linguistic tasks. Considering the comprehensive nature of their training datasets, which span a wide variety of knowledge domains, the expert knowledge embedded in LLMs can be employed to provide class labels visually similar to the ID class to meet our needs.

**Zero-Shot OOD Detection** (Esmaeilpour et al., 2022; Ming et al., 2022; Wang et al., 2023) enables detecting OOD samples across diverse ID datasets using the same pre-trained model, *e.g.*, CLIP, without re-training on any unseen ID data. It can be viewed as a binary classification problem:

$$G_\lambda(x; \mathcal{Y}_{\text{id}}, \mathcal{I}, \mathcal{T}) = \begin{cases} \text{ID} & S(x) \geq \lambda \\ \text{OOD} & S(x) < \lambda \end{cases}, \tag{1}$$

where $G(\cdot)$ is the OOD detector, $x$ denotes the input image, $x \in \mathcal{X}$, $\mathcal{X} := \{\text{ID}, \text{OOD}\}$ and $\mathcal{Y}_{\text{id}}$ defines the space of ID class labels. The OOD detection score function $S$ is derived from the similarity between the visual representation $\mathcal{I}(x)$ and textual representation $\mathcal{T}(t)$. $t$ is the textual input to the text encoder, such as *"a photo of a <ID class>"*. $\lambda$ is the threshold to distinguish ID/OOD classes.

## 3 DREAMING OUTLIER SEMANTICS FOR ZERO-SHOT OOD DETECTION

In this paper, we aim to enhance zero-shot OOD detection performance by harnessing LLM to generate potential outlier class labels. However, several challenges require attention: (1) How to guide LLM to generate the desired outlier class label? (2) How can we sharpen the distinction between ID and OOD samples given the dreamed outlier class label? To address these issues, we propose LLM prompts specifically tailored for OOD detection and introduce a novel score function for better differentiation. The overall framework of our method is illustrated in Figure. 2.

## 3.1 Acquiring Dreamed Outlier Class Labels For OOD Detection

We first categorize OOD detection tasks into three types: Far, Near, and Fine-grained OOD Detection. Then, we elaborate the corresponding three LLM prompts anchored on *visual similarity* to produce outlier class labels as follows, which are general to different datasets for each OOD task, respectively.

**Far OOD Detection** refers to identifying OOD classes that are distant from the ID classes in the label space, with most being effectively discerned. Building upon the existing ID class labels, we first guide LLM to summarize these classes and determine their respective major categories. Subsequently, we ask LLM to provide outlier class labels that are visually similar to these major categories. Since LLM first summarizes the ID classes into major categories, our approach can be easily extended to large-scale datasets, such as ImageNet-1K. The LLM prompt for far OOD detection is illustrated in Figure 3. Wherein, the $\mathcal{Y}_{\text{id}}$ represents the set of ID class labels, $\mathcal{Y}_{\text{id}} := \{y_1, y_2, \cdots, y_K\}$, and the $K$ signifies the total number of categories encompassed within the ID class labels. Similarly, $\mathcal{Y}_{\text{dood}}$ indicates the set of dreamed outlier class labels generated by LLM, and $\mathcal{Y}_{\text{dood}} := \{n_1, n_2, \cdots, n_L\}$.

> **Q:** I have gathered images of $K$ distinct categories: $\mathcal{Y}_{\text{id}}$. Summarize what broad categories these categories might fall into based on visual features. Now, I am looking to identify $L$ classes that visually resemble these broad categories but have no direct relation to these broad categories. Please list these $L$ categories for me.
>
> **Far OOD prompt**                 **A:** These $L$ categories are:

Figure 3: LLM prompt for far OOD detection, consisting of both the contents of **Q** and **A**.

**Near OOD Detection** pertains to identifying OOD classes that are relatively close to the ID class, *e.g.*, *horse* and *zebra*, presenting an increased propensity to come across OOD samples that bear visual resemblances to ID classes. Consequently, for each ID class label, we instruct LLM to provide $l$ outlier class labels that exhibit visual resemblance with ID class labels, $l \times K = L$. Overlapping classes in $\mathcal{Y}_{\text{dood}}$ with $\mathcal{Y}_{\text{id}}$ are removed by string matching. Figure 4 illustrates the LLM prompt here.

> **Q:** Given the image category $y_i$, please suggest visually similar categories that are not directly related or belong to the same primary group as $y_i$. Provide suggestions that share visual characteristics but are from broader and different domains than $y_i$.
>
> **Near OOD prompt**          **A:** There are $l$ classes similar to $y_i$, and they are from broader and different domains than $y_i$:

Figure 4: LLM prompt for near OOD detection.

**Fine-grained OOD Detection**, also known as open-set recognition (OSR) (Vaze et al., 2021), focuses on semantic shift instead of distributional shift primarily comprised in traditional OOD detection. In fine-grained OOD detection, both ID and OOD samples fall under the same major category (*e.g.*, bird), and intrinsic visual similarities exist among subclasses (*e.g.*, Frigatebird, Ovenbird). Therefore, it is more appropriate to instruct the LLM to provide different subclasses within the same major category directly. The LLM prompt for fine-grained OOD detection is presented below Figure 5, where *class-type* refers to the major category, such as "*bird*".

> **Q:** I have a dataset containing $K$ different species of *class-type*. I need a list of $L$ distinct *class-type* species that are NOT present in my dataset, and ensure there are no repetitions in the list you provide. For context, the species in my dataset are: $\mathcal{Y}_{\text{id}}$.
>
> **Fine-grained OOD prompt**      **A:** The other $L$ *class-type* species not in the dataset are:

Figure 5: LLM prompt for fine-grained OOD Detection.

For detailed LLM prompts and the outlier class labels generated by LLM, please refer to Appendix. E.

## 3.2 A New OOD Detector Score

Given the dreamed outlier class labels, a straightforward strategy would be to classify input samples that exhibit the highest similarity on the outlier classes as OOD samples. Yet, this risks misidentifying an excessive number of ID inputs as OOD, resulting in decreased performance. To solve this problem, we introduce a new score function $S_{\text{DOS}}$ based on the proportionality between $\mathcal{Y}_{\text{dood}}$ and $\mathcal{Y}_{\text{id}}$ to distinguish hard OOD samples effectively. First, the label-wise matching score $s_i(x)$ is expressed as:

$$s_i(x) = \frac{\mathcal{I}(x) \cdot \mathcal{T}(t_i)}{\|\mathcal{I}(x)\| \cdot \|\mathcal{T}(t_i)\|}; \quad t_i \in \mathcal{D}_{\text{id}} \cup \mathcal{D}_{\text{dood}}. \tag{2}$$

Subsequently, taking into account the outlier class labels, the proposed score $S_{\text{DOS}}(\cdot)$ is defined as:

$$S_{\text{DOS}}(x; \mathcal{Y}_{\text{id}}, \mathcal{Y}_{\text{dood}}, \mathcal{T}, \mathcal{I}) = \max_{i \in [1,K]} \frac{e^{s_i(x)/\tau}}{\sum_{j=1}^{K+L} e^{s_j(x)/\tau}} - \max_{k \in [K+1, K+L]} \frac{\beta e^{s_k(x)/\tau}}{\sum_{j=1}^{K+L} e^{s_j(x)/\tau}}, \tag{3}$$

where $\tau$ is the temperature. $\beta$ indicates the proportion between outlier and ID class labels, formulated as $\beta = \frac{K}{K+L}$. For more details on the design of $S_{\text{DOS}}$, please refer to the Appendix. A. Based on $S_{\text{DOS}}$, the OOD detector $G(x; \mathcal{Y}_{\text{id}}, \mathcal{I}, \mathcal{T})$ can be viewed as the binary classification as:

$$G_\lambda(x; \mathcal{Y}_{\text{id}}, \mathcal{Y}_{\text{dood}}, \mathcal{T}, \mathcal{I}) = \begin{cases} \text{ID} & S_{\text{DOS}}(x) \geq \lambda \\ \text{OOD} & S_{\text{DOS}}(x) < \lambda \end{cases}, \tag{4}$$

where $\lambda$ is a selected threshold such that a high fraction of ID data (typically 95%) exceeds this value.

We summarize the advantages of our approach as follows:

1. **OOD-Agnostic**: DOS does not rely on prior knowledge of unknown OOD data, making it particularly suitable and adaptable to open-world scenarios.

2. **Zero-Shot**: A single pre-trained model efficiently serves various task-specific ID datasets without the need for individual training on each specific ID dataset. DOS can achieve superior OOD detection performance by merely knowing the ID class labels.

3. **Scalability and Generalizability**: In contrast to the existing zero-shot OOD detection method (Esmaeilpour et al., 2022) that generates candidate OOD class labels, DOS can be easily applied to large-scale datasets like ImageNet-1K. Moreover, DOS exhibits generalizability across diverse tasks, including far, near, and fine-grained OOD detection.

## 4 Experiments

### 4.1 Setups

**Far OOD Detection.** The ID datasets for far OOD detection encompass CUB-200-2011 (Wah et al., 2011), STANFORD-CARS (Krause et al., 2013), Food-101 (Bossard et al., 2014), Oxford-IIIT Pet (Parkhi et al., 2012) and ImageNet-1K (Deng et al., 2009). As for the OOD datasets, we use the large-scale OOD datasets iNaturalist (Van Horn et al., 2018), SUN (Xiao et al., 2010), Places (Zhou et al., 2017), and Texture (Cimpoi et al., 2014) curated by MOS (Huang & Li, 2021).

**Near OOD Detection.** We adopt ImageNet-10 and ImageNet-20 alternately as ID and OOD datasets, proposed by MCM (Ming et al., 2022), both of which are subsets extracted from ImageNet-1K. The ImageNet-10 dataset curated by MCM mimics the class distribution of CIFAR-10 (Krizhevsky et al., 2009). The ImageNet-20 dataset consists of 20 classes semantically similar to ImageNet-10 (e.g., dog (ID) vs. wolf (OOD)).

**Fine-grained OOD Detection.** We split CUB-200-2011, STANFORD-CARS, Food-101, and Oxford-IIIT Pet. Specifically, half of the classes from each dataset are randomly selected as ID data, while the remaining classes constitute OOD data. Importantly, there is no overlap between the above ID dataset and the corresponding OOD dataset.

**Evaluation Metrics.** We employ two widely-used metrics for evaluation: (1) FPR95, the false positive rate of OOD data when the true positive rate is at 95% for ID data, where a lower value

Table 1: Zero-shot **far** OOD detection results. The **bold** indicates the best performance on each dataset. The gray indicates that the comparative methods require an additional massive auxiliary dataset.

| ID Dataset | Method | OOD Dataset | | | | | | | | Average | |
| | | iNaturalist | | SUN | | Places | | Texture | | | |
| | | FPR95↓ | AUROC↑ | FPR95↓ | AUROC↑ | FPR95↓ | AUROC↑ | FPR95↓ | AUROC↑ | FPR95↓ | AUROC↑ |
|---|---|---|---|---|---|---|---|---|---|---|---|
| **CUB-200-2011** | CLIPN | 0.10 | 99.97 | 0.06 | 99.98 | 0.33 | 99.91 | 0.17 | 99.95 | 0.17 | 99.95 |
| | Energy | 0.46 | 99.89 | 0.03 | 99.99 | **0.30** | **99.92** | 0.02 | **100.00** | 0.20 | 99.95 |
| | MaxLogit | 0.35 | 99.92 | 0.06 | 99.99 | 0.35 | 99.91 | **0.00** | **100.00** | 0.19 | 99.95 |
| | MCM | 9.83 | 98.24 | 4.93 | 99.10 | 6.65 | 98.57 | 6.99 | 98.75 | 7.10 | 98.67 |
| | Ours | **0.07** | **99.98** | **0.02** | **100.00** | 0.33 | 99.90 | **0.00** | **100.00** | **0.10** | **99.97** |
| | GT | - | - | - | - | 0.29 | 99.93 | 0.00 | 99.99 | - | - |
| **STANFORD-CARS** | CLIPN | **0.00** | 99.99 | 0.02 | 99.99 | 0.13 | 99.96 | 0.02 | 99.99 | 0.04 | 99.98 |
| | Energy | 0.01 | **100.00** | 0.04 | 99.99 | 0.42 | 99.90 | 0.04 | 99.99 | 0.13 | 99.97 |
| | MaxLogit | **0.00** | **100.00** | 0.02 | 99.99 | 0.26 | 99.94 | **0.00** | **100.00** | 0.07 | 99.98 |
| | MCM | 0.05 | 99.77 | 0.02 | 99.95 | 0.24 | 99.89 | 0.02 | 99.96 | 0.08 | 99.89 |
| | Ours | **0.00** | 100.00 | **0.01** | 100.00 | **0.07** | **99.99** | **0.00** | 100.00 | **0.02** | **100.00** |
| | GT | - | - | - | - | 0.07 | 99.99 | 0.00 | 100.00 | - | - |
| **Food-101** | CLIPN | 0.70 | 99.83 | 0.10 | 99.96 | 0.26 | 99.94 | 5.35 | 98.19 | 1.60 | 99.48 |
| | Energy | 0.92 | 99.75 | 0.20 | 99.92 | 0.54 | 99.86 | 12.45 | 96.55 | 3.53 | 99.02 |
| | MaxLogit | 0.56 | 99.86 | 0.09 | 99.95 | 0.49 | 99.88 | 8.33 | 97.44 | 2.37 | 99.28 |
| | MCM | 0.64 | 99.78 | 0.90 | 99.75 | 1.86 | 99.58 | 4.04 | 98.62 | 1.86 | 99.43 |
| | Ours | **0.06** | **99.99** | **0.00** | **100.00** | **0.10** | **99.98** | **2.46** | **99.05** | **0.65** | **99.76** |
| | GT | - | - | - | - | 0.02 | 99.99 | 0.59 | 99.83 | - | - |
| **Oxford-IIIT Pet** | CLIPN | 0.01 | 99.99 | 1.08 | 99.78 | 0.97 | 99.80 | 1.42 | 99.61 | 0.87 | 99.80 |
| | Energy | 0.08 | 99.97 | 0.05 | 99.98 | 0.23 | 99.94 | 0.35 | 99.88 | 0.18 | 99.94 |
| | MaxLogit | 0.01 | 99.98 | 0.05 | 99.97 | 0.20 | 99.94 | 0.27 | 99.91 | 0.13 | 99.95 |
| | MCM | 2.80 | 99.38 | 1.05 | 99.73 | 2.11 | 99.56 | 0.80 | 99.81 | 1.69 | 99.62 |
| | Ours | **0.00** | **100.00** | **0.01** | **99.99** | **0.16** | **99.95** | **0.12** | **99.97** | **0.07** | **99.98** |
| | GT | - | - | - | - | 0.08 | 99.98 | 0.09 | 99.98 | - | - |
| **Average** | CLIPN | 0.20 | 99.95 | 0.32 | 99.93 | 0.42 | 99.90 | 1.74 | 99.44 | 0.67 | 99.80 |
| | Energy | 0.37 | 99.90 | 0.08 | 99.97 | 0.37 | 99.91 | 3.22 | 99.11 | 1.01 | 99.72 |
| | MaxLogit | 0.23 | 99.94 | 0.05 | 99.98 | 0.33 | 99.92 | 2.15 | 99.34 | 0.69 | 99.79 |
| | MCM | 3.33 | 99.29 | 1.72 | 99.63 | 2.72 | 99.40 | 2.96 | 99.29 | 2.68 | 99.40 |
| | Ours | **0.03** | **99.99** | **0.01** | **100.00** | **0.17** | **99.96** | **0.64** | **99.75** | **0.21** | **99.93** |
| | GT | - | - | - | - | 0.12 | 99.97 | 0.17 | 99.95 | - | - |

indicates better performance; (2) AUROC, the area under the receiver operating characteristic curve, with a higher value signifying superior performance.

**DOS Setups.** We employ CLIP (Ilharco et al., 2021) as the backbone of our framework. Specifically, the image encoder is selected from ViT-B/16 or ViT-L/14, with the latter having more parameters. Unless otherwise specified, we adopt ViT-B/16 as the image encoder and masked self-attention Transformer (Vaswani et al., 2017) as the text encoder in our experiments. The pre-trained weights of CLIP are sourced from the official weights provided by OpenAI. In addition, we adopt the GPT-3.5-turbo-16k model as the LLM for our research, setting the temperature parameter to 0. To reduce the potential impact of randomness, we instruct LLM to dream outlier class three times on each dataset independently, and the final reported results are the average of these three experiments. Throughout all our experiments, the temperature coefficient $\tau$ in score function $S_{\text{DOS}}$ is set to 1.

**Compared Methods.** We compare our method with state-of-the-art (SOTA) OOD detection methods, including zero-shot and those requiring fine-tuning. For fair comparisons, all compared methods employ CLIP as their backbone, consistent with DOS. With respect to fine-tuning methods, we consider MSP (Hendrycks & Gimpel, 2017), Energy (Liu et al., 2020), MOS (Huang & Li, 2021), and fort (Fort et al., 2021). As for zero-shot methods, our comparisons are drawn towards MCM (Ming et al., 2022) and CLIPN (Wang et al., 2023). What's more, We implement post-hoc methods (Energy (Liu et al., 2020) and MaxLogit (Hendrycks et al., 2019a)) as additional baselines on CLIP bacone. It is worth noting that CLIPN relies on a large-scale auxiliary dataset (Sharma et al., 2018) to additionally pre-train an encoder. Instead, our DOS does not require any such dataset.

## 4.2 MAIN RESULTS

**Far OOD Detection.** Table. 1 presents the comparison with the recent state-of-the-art zero-shot OOD detection method (MCM Ming et al. (2022)) across four ID datasets: CUB-200-2011, STANFORD-CARS, Food-50, and Oxford-IIIT Pet. For each dataset, we guide the LLM to dream 500 outlier classes, *i.e.*, $L = 500$. Clearly, DOS achieves superior results on these four ID datasets, **with an average FPR95 of 0.21% and AUROC of 99.93%**. This indicates substantial improvement over the strong baseline. Notably, our DOS outperforms the strong baseline by 7% when using CUB-200-2011 as the ID dataset.

Table 2: Zero-shot **far** OOD detection results for ImageNet-1K as ID dataset. The **black bold** indicates the best performance. The gray indicates that the comparative methods require training or an additional massive auxiliary dataset. Energy (FT) requires fine-tuning, while Energy is post-hoc.

| Method | OOD Dataset | | | | | | | | Average | |
| | iNaturalist | | SUN | | Places | | Texture | | | |
| | FPR95↓ | AUROC↑ | FPR95↓ | AUROC↑ | FPR95↓ | AUROC↑ | FPR95↓ | AUROC↑ | FPR95↓ | AUROC↑ |
|---|---|---|---|---|---|---|---|---|---|---|
| MOS (BiT) | 9.28 | 98.15 | 40.63 | 92.01 | 49.54 | 89.06 | 60.43 | 81.23 | 39.97 | 90.11 |
| Fort | 15.07 | 96.64 | 54.12 | 86.37 | 57.99 | 85.24 | 53.32 | 84.77 | 45.12 | 88.25 |
| Energy(FT) | 21.59 | 95.99 | 34.28 | 93.15 | 36.64 | 91.82 | 51.18 | 88.09 | 35.92 | 92.26 |
| MSP | 40.89 | 88.63 | 65.81 | 81.24 | 67.90 | 80.14 | 64.96 | 78.16 | 59.89 | 82.04 |
| CLIPN | 19.13 | 96.20 | 25.69 | 94.18 | 32.14 | 92.26 | 44.60 | 88.93 | 30.39 | 92.89 |
| Energy | 81.08 | 85.09 | 79.02 | 84.24 | 75.08 | 83.38 | 93.65 | 65.56 | 82.21 | 79.57 |
| MaxLogit | 61.63 | 89.32 | 64.36 | 87.43 | 63.70 | 85.95 | 86.67 | 71.69 | 69.09 | 83.60 |
| MCM | 30.91 | 94.61 | 37.59 | 92.57 | 44.69 | 89.77 | **57.77** | **86.11** | 42.74 | 90.77 |
| Ours | **7.10** | **98.32** | **16.72** | **96.49** | **26.48** | **94.00** | 69.57 | 80.65 | **29.97** | **92.37** |
| GT | - | - | - | - | 13.24 | 96.96 | 24.29 | 95.04 | - | - |

Table 3: Zero-shot **near** OOD Detection results. The **bold** indicates the best performance on each dataset, and the gray indicates methods requiring an additional massive auxiliary dataset.

| Method | ID OOD | ImageNet-10 ImageNet-20 | | ImageNet-20 ImageNet-10 | | Average | |
| | | FPR95↓ | AUROC↑ | FPR95↓ | AUROC↑ | FPR95↓ | AUROC↑ |
|---|---|---|---|---|---|---|---|
| CLIPN | | 7.80 | 98.07 | 13.67 | 97.47 | 10.74 | 97.77 |
| Energy | | 10.30 | 97.94 | 16.40 | 97.37 | 13.35 | 97.66 |
| MaxLogit | | 9.70 | 98.09 | **14.00** | 97.81 | 11.85 | 97.95 |
| MCM | | 5.00 | 98.71 | 17.40 | 97.87 | 11.20 | 98.29 |
| Ours | | **4.00** | **99.09** | 14.80 | **98.01** | **9.40** | **98.55** |
| GT | | 0.20 | 99.80 | 0.20 | 99.93 | 0.20 | 99.87 |

We then conduct experiments on the larger-scale dataset (ImageNet-1K) for far OOD detection. Results are reported in Table. 2. We adopt the results of fine-tuning methods reported by MCM. DOS is comparable to fine-tuning methods and surpasses MCM. Although CLIPN performs better than DOS on the ViT-B/16 backbone, such a comparison does not do justice to DOS. This is because CLIPN requires an additional text encoder and large-scale datasets for training the "no" prompt. Furthermore, CLIPN utilizes an ensemble strategy for the textual inputs, in which the ensemble and learnable textual inputs are effective in enhancing performance (Radford et al., 2021; Zhou et al., 2022b;a). We also explore the effectiveness of using a larger backbone (ViT-B/16) and different VLM (ALIGN (Jia et al., 2021)). When using ViT-B/16 or ALIGN as the backbone, our DOS's performance is significantly better than MCM. These results indicate that our DOS is more generalizable to different VLMs. Please refer to Table 5 in the Appendix. B for the details. What's more, we perform experiments with CIFAR-10/CIFAR-100 (Krizhevsky et al., 2009) benchmarks to support our methodology further. Please refer to the Appendix. C for the results.

**Near OOD Detection.** The results for near OOD detection are presented in Table. 3. For each ID class label, DOS instructs the LLM to return three outlier class labels. DOS outperforms the strong baseline MCM by achieving improvements of 1.80% in average FPR95 and 0.26% in AUROC. Compared to CLIPN, which uses extra large datasets for re-training, our DOS clearly outperforms it when using ImageNet-10 as the ID dataset and achieves competitive results with it when using ImageNet-20 as the ID dataset.

**Fine-grained OOD Detection.** Table. 4 shows the performance of fine-grained OOD detection. DOS guides the LLM to generate 500 outlier class labels for each ID dataset. Compared to MCM, our DOS largely increases the average OOD performance by 7.12% in FPR95 and 3.22% in AUROC. Despite the unfair comparison, our DOS still outperforms CLIPN on the CUB-100 and Food-50 datasets in terms of FPR95.

## 4.3 ABLATION STUDY

**Score Function.** To demonstrate the superiority of the proposed OOD detector score $S_{\text{DOS}}$, we compare it with the other score functions: $S_{\text{MAX}}, S_{\text{MSP}}, S_{\text{Eenergy}}$ and $S_{\text{MaxLogit}}$. Please refer to Appendix. D.1 for the specific forms of these score functions. The comparison of these score functions is shown in Figure. 6 (a). Results show that our $S_{\text{DOS}}$ achieves the best OOD performance. This verifies the superiority and importance of the proposed OOD detector score.

Table 4: Zero-shot **fine-grained** OOD Detection results. The **bold** indicates the best performance on each dataset, and the  gray  indicates methods requiring an additional massive auxiliary dataset.

| Method | ID OOD | CUB-100 CUB-100 | | Stanford-Cars-98 Stanford-Cars-98 | | Food-50 Food-51 | | Oxford-Pet-18 Oxford-Pet-19 | | Average | |
|---|---|---|---|---|---|---|---|---|---|---|---|
| | | FPR95↓ | AUROC↑ | FPR95↓ | AUROC↑ | FPR95↓ | AUROC↑ | FPR95↓ | AUROC↑ | FPR95↓ | AUROC↑ |
| CLIPN | | 73.54 | 74.65 | 53.33 | 82.25 | 43.33 | 88.89 | 53.90 | 86.92 | 56.05 | 83.18 |
| Energy | | 76.13 | 72.11 | 73.78 | 73.82 | 44.95 | 89.97 | 68.51 | 88.34 | 65.84 | 81.06 |
| MaxLogit | | 76.89 | 73.00 | **72.18** | **74.80** | 41.73 | 90.79 | 65.66 | 88.49 | 64.12 | **81.77** |
| MCM | | 83.58 | 67.51 | 83.99 | 68.71 | 43.48 | 91.75 | 63.92 | 84.88 | 68.72 | 78.21 |
| Ours | | **71.57** | **74.12** | 77.88 | 71.08 | **39.04** | **91.81** | **57.89** | **88.72** | **61.60** | 81.43 |
| GT | | 61.23 | 81.42 | 58.31 | 83.71 | 11.34 | 97.79 | 29.17 | 95.58 | 40.01 | 89.63 |

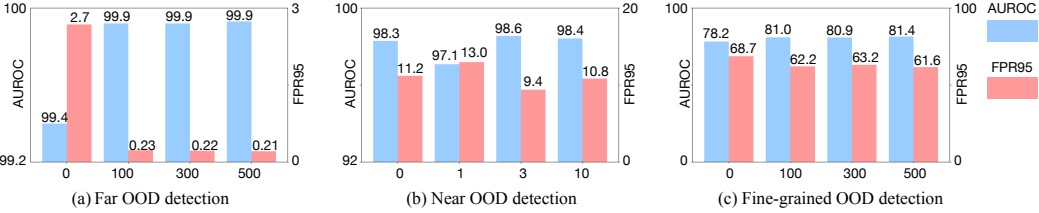

(a) Score Function  (b) LLM Prompts  (c) Various LLMs

Figure 6: Ablation study on (a) score function, (b) LLM prompts, and (c) various LLMs. ID dataset: ImageNet-10; OOD dataset: ImageNet-20.

(a) Far OOD detection  (b) Near OOD detection  (c) Fine-grained OOD detection

Figure 7: Evaluation on the number of outlier class labels. When the number of outlier class labels is zero, the method reduces to the baseline MCM.

**LLM Prompts.** To underscore the efficacy of *visually resemble*, we design two types of LLM prompts, one termed *visually irrelevant* and the other *visually dissimilar*, as well as maintaining the rest of the LLM prompts constant. Specifically, the *'irrelevant'* LLM prompt instructs the LLM to generate arbitrary outlier class labels for the ID class without adhering to a *visually resemble* constraint. Conversely, the *'dissimilar'* LLM prompt asks the LLM to dream outlier class labels for the ID class under a *visual dissimilar* constraint. As shown in Figure. 6 (b), without the *visually resemble* constraint proposed in our DOS, the OOD performance degrades on both FPR95 and AUROC metrics, indicating the importance of the proposed constraint. For detailed LLM prompts and the outlier class labels generated by LLM, please refer to Appendix. D.2.

**Various LLMs.** We conduct experiments with various LLMs to provide a more comprehensive understanding of DOS's effectiveness. Specifically, we use LLaMA2-7B or Claude2 to dream outlier class labels. The results on ImageNet10 (ID) are shown in Figure. 6 (c). All the variants, which use different LLMs, achieve better results than the baseline MCM. Moreover, Claude2 outperforms bo-16k in the FPR95 metric. These results demonstrate the universality of our method.

**Number of Outlier Class Labels.** We conduct experiments to investigate the impact of the number of outlier class labels, *i.e.*, $L$, generated by LLM. We instruct the LLM to return 100, 300, and 500 outlier class labels for the far and fine-grained OOD detection for each ID dataset. For the near OOD detection, we ask the LLM to return outlier class labels in quantities of 1, 3, and 10 for each ID class. Figure. 7 presents the respective average metrics for different numbers of outlier class labels within the three tasks. In near OOD detection, a limited number of outlier class labels (*e.g.*, 1) leads to a significant decline in performance. This is attributed to an increased propensity to encounter hard samples in near OOD detection. Although $L$ influences DOS performance, DOS can consistently outperform the baseline MCM and achieve relatively stable performance when $L$ is not too small.

## 4.4 VISUALIZATION

To better understand our DOS, we display the visualizations derived from the softmax output for the label-wise matching score via T-SNE (Van Der Maaten, 2014). Results compared between our DOS

and the baseline MCM are shown in Figure. 8. We can find that MCM tends to cluster all the OOD samples together. In our DOS, 1) OOD samples belonging to the same class tend to be clustered together and 2) samples in the same group are classified into the dreamed outlier class that is visually similar to them (Steam Locomotive *vs* Submarine). These observations indicate that our DOS is more semantically explainable.

## 5 RELATED WORKS

**Traditional OOD Detection.** Methods adopted for traditional OOD detection can be broadly categorized into classification-based, density-based (Ren et al., 2019; Xiao et al., 2020), and reconstruction-based (Denouden et al., 2018; Zhou, 2022; Liu et al., 2023). The classification-based methods leverage a well-trained ID classifier and formulate a scoring function to recognize OOD samples. The score function can be formulated from input (Perera et al., 2020), hidden (Sun et al., 2022; Lee et al., 2018; Sun & Li, 2022), output (Hendrycks & Gimpel, 2017; Liu et al., 2020), and gradient space (Huang et al., 2021). When the label space of the test ID data differs from that of the training data, the model needs to be re-trained from scratch or fine-tuned in traditional OOD detection scenarios, which require significant computational overhead.

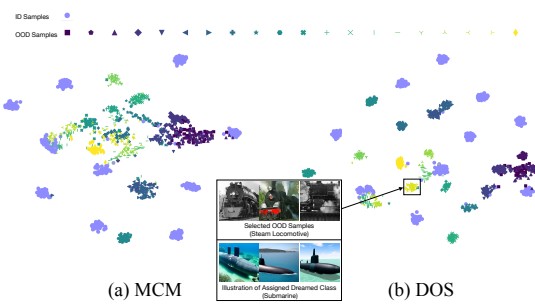

(a) MCM       (b) DOS

Figure 8: T-SNE visualizations obtained by the classifier output. ID set: ImageNet-10; OOD set: ImageNet-20. We use distinct colors to represent different OOD classes. The illustrated dreamed OOD name is the class assigned with the corresponding cluster, and its examples are generated by Stable Diffusion (Rombach et al., 2022). Best viewed with zoom in.

**Zero-Shot OOD Detection.** Owing to the powerful capabilities of the VLMs, zero-shot OOD detection methods have shown promising results. Typically, the similarity between the feature representations of input images and the textual input is measured to identify OOD samples. MCM (Ming et al., 2022) is entirely dependent on closed-set ID class labels and does not effectively harness the potent abilities of CLIP for handling open-world inputs. Although ZOC (Esmaeilpour et al., 2022) and CLIPN (Wang et al., 2023) take into account the open-world setting, ZOC requires an additional auxiliary dataset to train a text-based image description generator to generate candidate unknown class names for each input sample. This makes ZOC ill-suited for handling large-scale datasets. Similarly, CLIPN requires an extra auxiliary dataset for training the text encoder. By contrast, DOS not only considers the open-world scenario but also foregoes the need for any auxiliary datasets for extra training and can be easily scaled to large datasets.

**LLM for Visual Classification.** Drawing upon the expert knowledge embedded in LLMs has emerged as a novel trend in vision tasks and remains under-explored. Menon & Vondrick (2023); Maniparambil et al. (2023) employ the expertise within LLMs to extract the inherent information contained in ID class labels, thereby enhancing the performance of image classification (classification by description). Differing from this, DOS leverages the expert knowledge in LLMs to dream outlier OOD classes based on the visual similarity rule, harnessing the capabilities of VLMs more effectively, thus improving the performance of identifying OOD samples.

## 6 CONCLUSION

In this paper, we propose a new paradigm for zero-shot OOD detection, called DOS, by harnessing the expert knowledge embedded in LLMs to dream outlier semantics without relying on actual or auxiliary OOD data. Based on the designed visual similarity rule, the proposed three LLM prompts are applicable across various datasets for far, near, and fine-grained OOD detection tasks. We introduce a new score function based on the proportionality between potential outlier and ID class labels, enabling us to recognize OOD samples effectively. Extensive experiments show that DOS achieves new state-of-the-art performance and can be effectively scaled to the large-scale ImageNet-1K dataset. We hope this work could open a new door in future research in the OOD detection field.

## ETHIC STATEMENT

This paper does not raise any ethical concerns. This study does not involve any human subjects, practices to data set releases, potentially harmful insights, methodologies and applications, potential conflicts of interest and sponsorship, discrimination/bias/fairness concerns, privacy and security issues, legal compliance, and research integrity issues.

## REPRODUCIBILITY STATEMENT

The experimental setups for evaluation are described in detail in Section 4, and the experiments are all conducted using public datasets. The detailed LLM prompts and outlier class labels generated by LLM are provided in Appendix. E. We will release the code upon publication.

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

## A  SCORE FUNCTION DETAILS

The design of our score function $S_{\text{DOS}}$ jointly considers two key aspects: the utilization of dreamed outlier class labels and the balancing of their quantity. As for the former, the intuitive idea is to incorporate the dreamed class into the denominator of the MSP score (i.e., $S_{\text{MSP}}(x) = \max_{i \in [1,K]} \frac{e^{s_i(x)/\tau}}{\sum_{j=1}^{K+L} e^{s_j(x)/\tau}}$). However, in this case, the dreamed class only functions in the denominator, which doesn't significantly impact the final score distribution, implying that the dreamed class is not fully utilized. To amplify the role of the dream class, DOS further subtracts the second item$(-\max_{k \in [K+1,K+L]} \frac{\beta e^{s_k(x)/\tau}}{\sum_{j=1}^{K+L} e^{s_j(x)/\tau}})$. It stems from an intuition: samples visually similar to the dreamed class should have lower scores, thus making it easier to distinguish between the ID and OOD score distribution.

As for the latter, when the number of dreamed outlier class labels $L$ is large, the dreamed outlier class, merely being in the denominator, already significantly influences the overall score distribution. If we set $\beta = 1$, this may lead to the dreamed outlier class overly influencing the score distribution, potentially causing a decline in performance. Therefore, it is reasonable to adaptively adjust $\beta$ based on the size of $L$. We thus design $\beta = \frac{K}{K+L}$, ensuring a balanced influence of the dreamed outlier class.

## B  DIFFERENT VLM BACKBONES

Table. 5 shows the performance of ImageNet-1K(ID) on ViT-B/16 and ALIGN. Compared with ViT-B/16, DOS yields an enhancement of 2.16% and 0.64% in FPR95 and AUROC based on ViT-L/14, respectively. Moreover, DOC achieves the best OOD detection performance compared to both zero-shot methods and fine-tuning methods in terms of FPR95. It should be noted that CLIPN achieves worse performance when using the VIT-L/14 as the backbone than when using VIT-B/16. Instead, our DOS is more generalizable to different backbones and produces clearly better OOD performance than CLIPN when using VIT-L/14 as the backbone. When using the ALIGN backbone, our DOS (28.73%) improves the FPR95 by 29.90% compared to the baseline MCM (58.63%).

Table 5: Zero-shot **far** OOD detection results for ImageNet-1K as ID dataset. The **black bold** indicates the best performance based on CLIP (ViT-L/14), and the **blue bold** signifies the best performance using ALIGN. The gray indicates that the comparative methods require training or an additional massive auxiliary dataset. Energy (FT) requires fine-tuning, while Energy is post-hoc.

| Method | OOD Dataset | | | | | | | | | |
| | iNaturalist | | SUN | | Places | | Texture | | Average | |
| | FPR95↓ | AUROC↑ | FPR95↓ | AUROC↑ | FPR95↓ | AUROC↑ | FPR95↓ | AUROC↑ | FPR95↓ | AUROC↑ |
|---|---|---|---|---|---|---|---|---|---|---|
| MOS (BiT) | 9.28 | 98.15 | 40.63 | 92.01 | 49.54 | 89.06 | 60.43 | 81.23 | 39.97 | 90.11 |
| Fort (ViT-L/14) | 15.74 | 96.51 | 52.34 | 87.32 | 55.14 | 86.48 | 51.38 | 85.54 | 43.65 | 88.96 |
| Energy (ViT-L/14) | 10.62 | 97.52 | 30.46 | 93.83 | 32.25 | 93.01 | 44.35 | 89.64 | 29.42 | 93.50 |
| MSP (ViT-L/14) | 34.54 | 92.62 | 61.18 | 83.68 | 59.86 | 84.10 | 59.27 | 82.31 | 53.71 | 85.68 |
| CLIPN (ViT-L/14) | 25.09 | 94.59 | 24.76 | 94.93 | 30.89 | 93.14 | 48.97 | 87.01 | 32.43 | 92.42 |
| Energy (ViT-L/14) | 78.84 | 85.87 | 78.87 | 83.51 | 70.30 | 86.44 | 93.90 | 63.74 | 80.48 | 79.89 |
| MaxLogit (ViT-L/14) | 58.91 | 90.13 | 63.91 | 87.75 | 56.98 | 89.05 | 88.85 | 69.37 | 67.16 | 84.08 |
| MCM (ViT-L/14) | 28.38 | 94.95 | 29.00 | 94.14 | 35.42 | 92.00 | **59.88** | **84.88** | 38.17 | 91.49 |
| Ours (ViT-L/14) | **6.54** | **98.45** | **14.52** | **97.10** | **22.28** | **94.94** | 67.89 | 81.51 | **27.81** | **93.01** |
| Energy (ALIGN) | 92.21 | 83.62 | 74.16 | 86.29 | 70.50 | 84.65 | 68.60 | 81.77 | 76.37 | 84.08 |
| MaxLogit (ALIGN) | 83.46 | 84.95 | 68.46 | 86.63 | 67.10 | 84.78 | 65.32 | 81.90 | 71.08 | 84.56 |
| MCM (ALIGN) | 60.63 | 89.39 | 53.20 | 89.17 | 61.05 | 85.58 | **59.63** | **83.94** | 58.63 | 87.02 |
| Ours (ALIGN) | **9.54** | **98.07** | **15.38** | **96.77** | **25.24** | **93.89** | 64.75 | 81.57 | **28.73** | **92.57** |

## C  OTHER OOD DETECTION BENCHMARKS

We conduct experiments on the CIFAR-10/CIFAR-100 (SVHN, LSUN, Texture, Places) datasets, and the results are shown in Table. 6. Clearly, DOS achieves superior results on this standard OOD detection benchmark.

## D  ABLATION STUDY

### D.1  SCORE FUNCTION

Table 6: Zero-shot OOD detection results in cifar10 and cifar100 benchmarks. The **bold** indicates the best performance on each dataset. The gray indicates that the comparative methods require training or an additional massive auxiliary dataset.

| ID Dataset | Method | SVHN | | LSUN | | Texture | | Places | | Average | |
|---|---|---|---|---|---|---|---|---|---|---|---|
| | | FPR95↓ | AUROC↑ | FPR95↓ | AUROC↑ | FPR95↓ | AUROC↑ | FPR95↓ | AUROC↑ | FPR95↓ | AUROC↑ |
| | CLIPN | 53.28 | 74.20 | 27.89 | 92.72 | 3.58 | 98.93 | 9.82 | 96.98 | 23.64 | 90.63 |
| | Energy | 18.97 | 96.67 | 60.60 | 88.81 | 16.13 | 96.59 | 17.48 | 95.42 | 28.29 | 94.37 |
| cifar10 | MaxLogit | 6.50 | 98.27 | 36.54 | 94.20 | 11.37 | 97.61 | 16.67 | 95.56 | 17.77 | 96.41 |
| | MCM | **3.98** | **99.03** | 5.12 | 98.72 | 16.35 | 96.44 | 36.55 | 90.79 | 15.50 | 96.25 |
| | Ours | 4.48 | 98.51 | **3.24** | **98.88** | **6.91** | **98.32** | 15.24 | 95.57 | **7.47** | **97.82** |
| | CLIPN | 71.72 | 68.20 | 84.42 | 80.90 | 37.74 | 90.92 | 51.06 | 87.25 | 61.24 | 81.82 |
| | Energy | 72.54 | 88.20 | 93.64 | 73.08 | 65.55 | 80.43 | 59.86 | **83.47** | 72.90 | 81.30 |
| cifar100 | MaxLogit | 59.05 | **91.01** | 82.48 | 83.06 | 62.82 | 82.08 | 65.58 | 80.88 | 67.48 | 84.26 |
| | MCM | 64.45 | 89.96 | 47.26 | 91.69 | 90.30 | 73.61 | 98.42 | 61.37 | 75.11 | 79.16 |
| | Ours | 75.64 | 87.00 | **39.04** | **92.87** | **62.00** | 83.54 | 70.60 | 80.43 | **61.82** | **85.95** |
| | CLIPN | 62.50 | 71.20 | 56.16 | 86.81 | 20.66 | 94.93 | 30.44 | 92.12 | 42.44 | 86.23 |
| | Energy | 45.76 | 92.44 | 77.12 | 80.95 | 40.84 | 88.51 | **38.67** | **89.45** | 50.60 | 87.84 |
| Average | MaxLogit | 32.78 | **94.64** | 59.51 | 88.63 | 37.10 | 89.85 | 41.13 | 88.22 | 42.63 | 90.34 |
| | MCM | 34.22 | 94.50 | 26.19 | 95.21 | 53.33 | 85.03 | 67.49 | 76.08 | 45.31 | 87.71 |
| | Ours | 40.06 | 92.76 | **21.14** | **95.88** | **34.46** | **90.93** | 42.92 | 88.00 | **34.65** | **91.88** |

Here, we present the specific form of the score function designed in the ablation study:

$$S_{\text{MAX}}(x; \mathcal{Y}_{\text{id}}, \mathcal{Y}_{\text{dood}}, \mathcal{T}, \mathcal{I}) = \begin{cases} \frac{1}{K} & \max_{i \in [1,K]} s_i < \max_{j \in [K+1,L]} s_j \\ \max_{i \in [1,K]} \frac{e^{s_i(x)/\tau}}{\sum_{j=1}^{K} e^{s_j(x)/\tau}} & \max_{i \in [1,K]} s_i \geq \max_{j \in [K+1,L]} s_j \end{cases}, \quad (5)$$

$$S_{\text{MSP}}(x; \mathcal{Y}_{\text{id}}, \mathcal{Y}_{\text{dood}}, \mathcal{T}, \mathcal{I}) = \max_{i \in [1,K]} \frac{e^{s_i(x)/\tau}}{\sum_{j=1}^{K+L} e^{s_j(x)/\tau}}, \quad (6)$$

$$S_{\text{Energy}}(x; \mathcal{Y}_{\text{id}}, \mathcal{Y}_{\text{dood}}, \mathcal{T}, \mathcal{I}) = -T \left( \log \sum_{i=1}^{K} e^{f_i(x)/T} - \log \sum_{j=K+1}^{L} e^{f_j(x)/T} \right), \quad (7)$$

$$S_{\text{MaxLogit}}(x; \mathcal{Y}_{\text{id}}, \mathcal{Y}_{\text{dood}}, \mathcal{T}, \mathcal{I}) = \max_{i \in [1,K]} s_i(x) - \max_{j \in [K+1,K+L]} s_j(x). \quad (8)$$

$S_{\text{MAX}}$ indicates that if the $s_j$ of an input sample is greater than the $s_i$, this sample is recognized to be an OOD sample. Otherwise, the input sample is calculated according to maximum softmax probabilities (MSP). $s_j$ and $s_i$ represent the label-wise matching scores as defined in Eq. 2. $S_{\text{MSP}}$ is an adaptation of MSP, integrating outlier classes into its denominator. Similarly, $S_{\text{Energy}}$ and $S_{\text{MaxLogit}}$ are modifications of the Energy and MaxLogit metrics, respectively, incorporating outlier classes into their secondary components.

## D.2 LLM PROMPTS

We provide specific examples of *'dissimilar'* and *'irrelevant'* LLM prompts in Figure. 12 and Figure. 9, respectively. The LLM chosen is GPT-3.5-turbo-16k, accessed on September 20, 2023.

## E SPECIFIC EXAMPLE OF LLM PROMPTS

To obtain the outlier class labels automatically generated by the LLM, we provide a question and answer template for LLM and append '-' to the end of the answer to produce a bullet-point list output. It is worth noting that the provided template does not contain any ID class content. Below, we present specific examples for three LLM prompts. The LLM prompt for far OOD detection is depicted in Figure. 10. The LLM prompts for near OOD detection and fine-grained OOD detection are showcased in Figure. 11 and Figure. 13, respectively. We employed the GPT-3.5-turbo-16k LLM, accessed on September 20, 2023.

**Q:** Given the image category [water jug], please suggest categories that are not directly related or belong to the same primary group as [water jug].
**A:** There are three classes from broader and different domains than [water jug]:
- trumpets
- helmets
- rucksacks

**Q:** Given the image category [horse], please suggest categories that are not directly related or belong to the same primary group as [horse].
**A:** There are three classes from broader and different domains than [horse]:

ID class label: horse

- pineapple
- laptop
- sunglasses

Figure 9: Instance of *'irrelevant'* LLM prompt for OOD detection, ID class label: *horse*. Note that, the gray is the LLM prompt for near OOD detection, and the green is the LLM actually returns.

**Q:** I have gathered images of 4 distinct categories: ['Husky dog', 'Garfield cat', 'churches', 'truck']. Summarize what broad categories these categories might fall into based on visual features. Now, I am looking to identify 5 categories that visually resemble these broad categories but have no direct relation to these broad categories. Please list these 5 items for me.
**A:** These 5 items are:
- black stone
- mountain
- Ginkgo Tree
- river
- Rapeseed

**Q:** I have gathered images of 100 distinct categories: ['Apple pie', 'Baby back ribs', 'Baklava', 'Beef carpaccio', 'Beef tartare', 'Beet salad', 'Beignets', 'Bibimbap', 'Bread pudding', 'Breakfast burrito', 'Bruschetta', 'Caesar salad', 'Cannoli', 'Caprese salad', 'Carrot cake', 'Ceviche', 'Cheesecake', 'Cheese plate', 'Chicken curry', 'Chicken quesadilla', 'Chicken wings', 'Chocolate cake', 'Chocolate mousse', 'Churros', 'Clam chowder', 'Club sandwich', 'Crab cakes', 'Creme brulee', 'Croque madame', 'Cup cakes', 'Deviled eggs', 'Donuts', 'Dumplings', 'Edamame', 'Eggs benedict', 'Escargots', 'Falafel', 'Filet mignon', 'Fish and chips', 'Foie gras', 'French fries', 'French onion soup', 'French toast', 'Fried calamari', 'Fried rice', 'Frozen yogurt', 'Garlic bread', 'Gnocchi', 'Greek salad', 'Grilled cheese sandwich', 'Grilled salmon', 'Guacamole', 'Gyoza', 'Hamburger', 'Hot and sour soup', 'Hot dog', 'Huevos rancheros', 'Hummus', 'Ice cream', 'Lasagna', 'Lobster bisque', 'Lobster roll sandwich', 'Macaroni and cheese', 'Macarons', 'Miso soup', 'Mussels', 'Nachos', 'Omelette', 'Onion rings', 'Oysters', 'Pad thai', 'Paella', 'Pancakes', 'Panna cotta', 'Peking duck', 'Pho', 'Pizza', 'Pork chop', 'Poutine', 'Prime rib', 'Pulled pork sandwich', 'Ramen', 'Ravioli', 'Red velvet cake', 'Risotto', 'Samosa', 'Sashimi', 'Scallops', 'Seaweed salad', 'Shrimp and grits', 'Spaghetti bolognese', 'Spaghetti carbonara', 'Spring rolls', 'Steak', 'Strawberry shortcake', 'Sushi', 'Tacos', 'Takoyaki', 'Tiramisu', 'Tuna tartare', 'Waffles']. Summarize what broad categories these categories might fall into based on visual features. Now, I am looking to identify 50 classes that visually resemble these broad categories but have no direct relation to these broad categories. Please list these 50 items for me.
**A:** These 50 items are:

ID dataset: Food-101

- Orchid
- Watermelon
- Sunflower
- ...
- Neon graffiti

Figure 10: Instance of LLM prompt for far OOD detection, ID dataset: Food-101. Note that, the gray is the LLM prompt for far OOD detection, and the green is the LLM actually returns.

Q: Given the image category [water jug], please suggest visually similar categories that are not directly related or belong to the same primary group as [water jug]. Provide suggestions that share visual characteristics but are from broader and different domains than [water jug].
A: There are three classes similar to [water jug], and they are from broader and different domains than [water jug]:
- trumpets
- helmets
- rucksacks

Q: Given the image category [horse], please suggest visually similar categories that are not directly related or belong to the same primary group as [horse]. Provide suggestions that share visual characteristics but are from broader and different domains than [horse].
A: There are three classes similar to [horse], and they are from broader and different domains than [horse]:

ID class label: horse

- zebra
- giraffe
- deer

Figure 11: Instance of LLM prompt for near OOD detection, ID class label: *horse*. Note that, the gray is the LLM prompt for near OOD detection, and the green is the LLM actually returns.

Q: Given the image category [water jug], please suggest visually dissimilar categories that are not directly related or belong to the same primary group as [water jug]. Provide suggestions that do not share visual characteristics but are from broader and different domains than [water jug].
A: There are three classes dissimilar to [water jug], and they are from broader and different domains than [water jug]:
- trumpets
- helmets
- rucksacks

Q: Given the image category [horse], please suggest visually dissimilar categories that are not directly related or belong to the same primary group as [horse]. Provide suggestions that do not share visual characteristics but are from broader and different domains than [horse].
A: There are three classes dissimilar to [horse], and they are from broader and different domains than [horse]:

ID class label: horse

- pineapple
- laptop
- mountain

Figure 12: Instance of *'dissimilar'* LLM prompt for OOD detection, ID class label: *horse*. Note that, the gray is the LLM prompt for near OOD detection, and the green is the LLM actually returns.

**Q:** I have a dataset containing 10 unique species of dogs. I need a list of 10 distinct dog species that are NOT present in my dataset, and ensure there are no repetitions in the list you provide. For context, the species in my dataset are: ['husky dog', 'alaskan Malamute', 'cossack sled dog', 'golden retriever', 'German Shepherd', 'Beagle', 'Bulldog', 'Poodle', 'Dachshund', 'Doberman Pinscher']
**A:** The other 10 dog species not in the dataset are:
- Labrador Retriever
- Rottweiler
- Boxer
- Border Collie
- Shih Tzu
- Akita
- Saint Bernard
- Australian Shepherd
- Great Dane
- Boston Terrier

**Q:** Q: I have a dataset containing 50 different species of food. I need a list of 50 distinct food species that are NOT present in my dataset, and ensure there are no repetitions in the list you provide. For context, the species in my dataset are: ['Prime Rib', 'Dumplings', 'Strawberry Shortcake', 'Frozen Yogurt', 'Seaweed Salad', 'Tiramisu', 'Red Velvet Cake', 'Omelette', 'Beef Carpaccio', 'Lasagna', 'Donuts', 'Sushi', 'Beignets', 'Chicken Wings', 'Carrot Cake', 'Gnocchi', 'Lobster Bisque', 'Spaghetti Bolognese', 'Greek Salad', 'Oysters', 'Caprese Salad', 'Panna Cotta', 'Shrimp And Grits', 'Baby Back Ribs', 'Creme Brulee', 'Gyoza', 'Escargots', 'Churros', 'Grilled Cheese Sandwich', 'Scallops', 'Breakfast Burrito', 'Cheesecake', 'Huevos Rancheros', 'Cheese Plate', 'Steak', 'Apple Pie', 'Mussels', 'Crab Cakes', 'Pancakes', 'Pulled Pork Sandwich', 'Bruschetta', 'Hot Dog', 'Risotto', 'Chicken Curry', 'Paella', 'Cannoli', 'Eggs Benedict', 'Fried Calamari', 'French Fries', 'Lobster Roll Sandwich']
**A:** The other 50 food species not in the dataset are:

ID dataset: Food50

- Lychee
- chicken tacos
- Durian
- ...
- affogato
- pizza

Figure 13: Instance of LLM prompt for fine-grained OOD detection, ID dataset: Food-50. Note that, the gray is the LLM prompt for fine-grained OOD detection, and the green is the LLM actually returns.

