# OpenReview forum: "DOS: Dreaming Outlier Semantics for Out-of-distribution Detection"
_ICLR.cc/2024/Conference — Submitted to ICLR 2024_

### Official Review · Reviewer_WwTG · 2023-10-16

**Soundness:** 2 fair
**Presentation:** 3 good
**Contribution:** 2 fair
**Rating:** 3
**Confidence:** 3

**Summary:**

This paper uses an LLM to generate potential OOD labels to be used alongside ID labels for score-matching with CLIP.

**Strengths:**

Simple method, clearly written and easy to understand.

Experiments are comprehensive with many datasets and ID/OOD setups.

Sufficient ablation study

**Weaknesses:**

Though the use of LLM to generate OOD labels is distinct from previous work, this alone does not seem like a strong novelty contribution compared to related works

The experimental setting choices are unclear. For example, only the MCM baseline is tested for Zero-shot far OOD with most datasets, but for ImageNet-1K there are many more baselines. Similar problem for near-OOD.These experiments should be more consistent.

There should be more discussion around how each dataset is adapted to be ID-OOD, For example, how similar are iNet-10 and iNet-20 really? It depends on the subsets of classes chosen.

The design of the LLM prompt causes some information leakage, in that a different type of prompt is used for far-, near- and fine-grained OOD settings. This means there is an implicit assumption about what type of anomalies are likely to be seen in a given experiment. A fairer method would not have such assumption about the test data embedded into its LLM prompts.

**Questions:**

Why use FPR95 instead of AUPR?

In Figure 7, why are far-ood and fine-grained-ood tested with 100s of outlier class labels but near-ood only with a maximum of 10?

---

> ### Author Response · Authors · 2023-11-18
> **Response to Reviewer WwTG (1/4)**
>
> We thank the reviewer WwTG for the valuable feedback. We addressed all the comments. Please find the point-to-point responses below. Any further comments and discussions are welcomed!
>
> >**Q1**. *Though the use of LLM to generate OOD labels is distinct from previous work, this alone does not seem like a strong novelty contribution compared to related works*
> >
> >**Reply:** We first would like to acknowledge that the reviewer identifies the diffrence between our method and previous methods. Moreover, we would like to clarify that our motivation is also new in the community. Our DOS is motivated by the observation that CLIP can achieve superior OOD detection performance when combined with ground truth OOD class labels. However, relying solely on the closed-set ID classes limits the original performance capabilities of CLIP. Since GT OOD class labels are unavailable, it is not easy to utlize candidate OOD names for solving the above issue. In this paper, we proposed employ LLM to dream outlier class labels to unleash  extraordinary OOD detection capability in CLIP.
> >
> >Based on the above motivation, we restate our novelty contribution as follows:
> >- We are the first to apply LLM to OOD detection task, and we have demonstrated the effectiveness of this approach. This provides a novel perspective for the OOD detection community.
> >- We design general LLM prompts specifically for OOD detection, which ask LLM to provide suitable dreamed OOD class labels.
> >- We design a new score function based on the proportionality between potential outlier class labels and ID class labels, helping the model effectively distinguish ID score distribution and OOD score distribution.
> >- In addition, our approach is generalized and not just limited to CLIP and GPT.
> >>- Our approach not only unleashes the potential of CLIP in open-world scenarios but also achieves similar objectives on other VLMs, such as ALIGN. For instance, on ImageNet-1K, using the ALIGN backbone, our DOS (28.73%) improves the FPR95 by 29.90% compared to the baseline MCM (58.63%). Experimental results are in the Appendix. B (Table. 5).
> >>- Our approach can also enhance OOD detection performance using other LLMs like LLaMA and Claude. Experimental results are in Section 4.3 (Figure. 6(c)).
> >- We conduct comprehensive experiments on various large-scale datasets to support our method.
> >>- Our DOS outperforms most approaches that require fine-tuning or additional dataset training.
> >>- We conducted sufficient ablation experiments to demonstrate the rationality of our design.

---

> ### Author Response · Authors · 2023-11-18
> **Response to Reviewer WwTG (2/4)**
>
> >**Q2**. *These experiments should be more consistent.*
> >
> >**Reply:** Thank you for your valuable feedback. Indeed, we followed the experimental setting of MCM[1]. In MCM, only its own performance was shown in far OOD task. On ImageNet dataset, we adopt the results of fine-tuning methods reported by MCM. After the work of MCM, only CLIPN[2] reported results on ImageNet-1K, so we did not compare with other other methods.
> >
> >To make experiments be consistent, we add CLIPN baseline in the far OOD task. Moreover, we implement the Energy[3] and MaxLogit[4] scores on the CLIP backbone as two additional comparative methods, which have been added to all of our experiments (Table. 1, 2, 3, 4).
>
> [1] Ming et al. Delving into Out-of-Distribution Detection with Vision-Language Representations. In NeurIPS, 2022.
>
> [2] Wang et al. CLIPN for Zero-Shot OOD Detection: Teaching CLIP to Say No. ICCV 2023.
>
> [3] Liu et al. Energy-based Out-of-distribution Detection. In NeurIPS, 2020.
>
> [4] Hendrycks et al. Scaling out-of-distribution detection for real-world settings. In ICML, 2022.
>
> Table 1: Far OOD, Due to characters limitations, Table 1 here only reports the average metrics for four ID datasets. For specific metrics of each ID dataset, please refer to Table 1 in the paper.
> | ID Dataset | Method | OOD Dataset |  |  |  |  |  |  |  | Average |  |
> | :---: | :---: | :---: | :---: | :---: | :---: | :---: | :---: | :---: | :---: | :---: | :---: |
> |  |  | iNaturalist |  | SUN |  | Places |  | Texture |  |  |  |
> |  |  | FPR95 $\downarrow$ | AUROC $\uparrow$ | FPR95 $\downarrow$ | AUROC $\uparrow$ | FPR95 $\downarrow$ | AUROC $\uparrow$ | FPR95 $\downarrow$ | AUROC $\uparrow$ | FPR95 $\downarrow$ | AUROC $\uparrow$ |
> | Average | CLIPN | 0.20 | 99.95 | 0.32 | 99.93 | 0.42 | 99.90 | 1.74 | 99.44 | 0.67 | 99.80 |
> |  | Energy | 0.37 | 99.90 | 0.08 | 99.97 | 0.37 | 99.91 | 3.22 | 99.11 | 1.01 | 99.72 |
> |  | MaxLogit | 0.23 | 99.94 | 0.05 | 99.98 | 0.33 | 99.92 | 2.15 | 99.34 | 0.69 | 99.79 |
> |  | MCM | 3.33 | 99.29 | 1.72 | 99.63 | 2.72 | 99.40 | 2.96 | 99.29 | 2.68 | 99.40 |
> |  | Ours | 0.03 | 99.99 | 0.01 | 100.00 | 0.17 | 99.96 | 0.64 | 99.75 | 0.21 | 99.93 |
>
>
> Table 2: ImageNet-1K(ID)
> | Method | OOD Dataset |  |  |  |  |  |  |  | Average |  |
> | :---: | :---: | :---: | :---: | :---: | :---: | :---: | :---: | :---: | :---: | :---: |
> |  | iNaturalist |  | SUN |  | Places |  | Texture |  |  |  |
> |  | FPR95 $\downarrow$ | AUROC $\uparrow$ | FPR95 $\downarrow$ | AUROC $\uparrow$  | FPR95 $\downarrow$ | AUROC $\uparrow$ | FPR95 $\downarrow$ | AUROC $\uparrow$ | FPR95 $\downarrow$ | AUROC $\uparrow$ |
> | CLIPN | 19.13 | 96.20 | 25.69 | 94.18 | 32.14 | 92.26 | 44.60 | 88.93 | 30.39 | 92.89 |
> | Energy | 81.08 | 85.09 | 79.02 | 84.24 | 75.08 | 83.38 | 93.65 | 65.56 | 82.21 | 79.57 |
> | MaxLogit | 61.63 | 89.32 | 64.36 | 87.43 | 63.70 | 85.95 | 86.67 | 71.69 | 69.09 | 83.60 |
> | MCM | 30.91 | 94.61 | 37.59 | 92.57 | 44.69 | 89.77 | 57.77 | 86.11 | 42.74 | 90.77 |
> | Ours | 7.10 | 98.32 | 16.72 | 96.49 | 26.48 | 94.00 | 69.57 | 80.65 | 29.97 | 92.37 |
>
> Table 3: Near OOD
> | Method | ID /OOD | ImageNet-10 / ImageNet-20 |  | ImageNet-20 / ImageNet-10 |  | Average |  |
> | :---: | :---: | :---: | :---: | :---: | :---: | :---: | :---: |
> |  |  | FPR95 $\downarrow$ | AUROC $\uparrow$  | FPR95 $\downarrow$ | AUROC $\uparrow$  | FPR95 $\downarrow$ | AUROC $\uparrow$ |
> | CLIPN |  | 7.80 | 98.07 | 13.67 | 97.47 | 10.74 | 97.77 |
> | Energy |  | 10.30 | 97.94 | 16.40 | 97.37 | 13.35 | 97.66 |
> | MaxLogit |  | 9.70 | 98.09 | 14.00 | 97.81 | 11.85 | 97.95 |
> | MCM |  | 5.00 | 98.71 | 17.40 | 97.87 | 11.20 | 98.29 |
> | Ours |  | 4.00 | 99.09 | 14.80 | 98.01 | 9.40 | 98.55 |
>
> Table 4: Fine-grained OOD
> | Method | ID / OOD | CUB-100 / CUB-100 |  | Stanford-Cars-98 /Stanford-Cars- 98 |  | Food-50 / Food-51 |  | Oxford-Pet-18 / Oxford-Pet-19 |  | Average |  |
> | :---: | :---: | :---: | :---: | :---: | :---: | :---: | :---: | :---: | :---: | :---: | :---: |
> |  |  | FPR95 $\downarrow$ | AUROC $\uparrow$  | FPR95 $\downarrow$ | AUROC $\uparrow$ | FPR95 $\downarrow$ |AUROC $\uparrow$ | FPR95 $\downarrow$ | AUROC $\uparrow$ | FPR95 $\downarrow$ | AUROC $\uparrow$ |
> | CLIPN |  | 73.54 | 74.65 | 53.33 | 82.25 | 43.33 | 88.89 | 53.90 | 86.92 | 56.05 | 83.18 |
> | Energy |  | 76.13 | 72.11 | 73.78 | 73.82 | 44.95 | 89.97 | 68.51 | 88.34 | 65.84 | 81.06 |
> | MaxLogit |  | 76.89 | 73.00 | 72.18 | 74.80 | 41.73 | 90.79 | 65.66 | 88.49 | 64.12 | 81.77 |
> | MCM |  | 83.58 | 67.51 | 83.99 | 68.71 | 43.48 | 91.75 | 63.92 | 84.88 | 68.72 | 78.21 |
> | Ours |  | 71.57 | 74.12 | 77.88 | 71.08 | 39.04 | 91.81 | 57.89 | 88.72 | 61.60 | 81.43 |

---

> ### Author Response · Authors · 2023-11-18
> **Response to Reviewer WwTG (3/4)**
>
> >**Q3**. *There should be more discussion around how each dataset is adapted to be ID-OOD, For example, how similar are iNet-10 and iNet-20 really? It depends on the subsets of classes chosen.*
> >
> >**Reply:** Thank you for your valuable feedback.
> >- In the far and near OOD tasks, we adhere to the setting of [MCM](https://github.com/deeplearning-wisc/MCM#Data-Preparation).
>     >>- The large-scale OOD datasets iNaturalist, SUN, Places, and Texture, curated by [[1]](https://github.com/deeplearning-wisc/large_scale_ood#out-of-distribution-dataset), are the standard benchmarks in the OOD detection area.
>    >> - The ImageNet-10 dataset curated by MCM mimics the class distribution of CIFAR-10. And the ImageNet-20 dataset consists of 20 classes semantically similar to ImageNet-10 (e.g., dog (ID) vs. wolf (OOD)).
> >- In the fine-grained task, We split CUB-200-2011, STANFORD-CARS, Food-101, and Oxford-IIIT Pet. Specifically, half of the classes from each dataset are randomly selected as ID data, while the remaining classes constitute OOD data.
> >
> >All the above ID and OOD datasets have been guaranteed to have no overlap. We have added the above discussion of dataset selection in the revised section 4.1.
>
> [1] Huang et al. MOS: Towards Scaling Out-of-distribution Detection for Large Semantic Space. In CVPR, 2021.
>
> >---
>
> >**Q4**. *The design of the LLM prompt causes some information leakage, in that a different type of prompt is used for far-, near- and fine-grained OOD settings. This means there is an implicit assumption about what type of anomalies are likely to be seen in a given experiment. A fairer method would not have such assumption about the test data embedded into its LLM prompts.*
> >
> >**Reply:** We first would like to emphasize is that there is no data leakage in our method, as we do not use the actual OOD data.
> >
> >Secondly, in OOD detection community, although there is no clear assumptation that we know the the types of anomalies in advance, existing approaches commonly evaluate different types (i.e., far and near OOD detection) individually. We thus follow this principle to design our prompts. Furthermore, in zero-shot OOD detection, all techniques necessitate prior knowledge of the ID class names that will be encountered during testing. We contend that possessing prior knowledge of the specific OOD task (far, near, or fine-grained) being assessed is analogous in information-level, and does not pose a risk of data leakage. On the other hand, experiments show that the proposed three prompts are universal and can be applied to various datasets for each task. This demonstrates its values in hanlding different scenarioes.
> >
> >Third, insipred by your comments, we conducted experiments without this prior knowledge, where we use the far prompt in all tasks. The experimental results, reported in the table bloew, show that our DOS still outperforms MCM in most cases. Admittedly, using different prompts is one minor drawback of our method. In this paper, we hope to propose a new perspective, using LLMs to explore potential OOD canditates, for solving the zero-shot OOD detection task. We would like to design a universal prompt in future work to further light this new direction.
>
>
> Table 1: using far OOD prompt in near OOD task
> | Method | ID / OOD | ImageNet-10 / ImageNet-20 |  | ImageNet-20 / ImageNet-10 |  | Average |  |
> | :---: | :---: | :---: | :---: | :---: | :---: | :---: | :---: |
> |  |  | FPR95 $\downarrow$ | AUROC $\uparrow$ | FPR95 $\downarrow$ | AUROC $\uparrow$ | FPR95 $\downarrow$ | AUROC $\uparrow$ |
> | MCM |  | 5.00 | 98.71 | 17.40 | 97.87 | 11.20 | 98.29 |
> | Ours |  |  4.00 | 99.09 | 14.80 | 98.01 | 9.40 | 98.55 |
> | Ours(far prompt) |  |  5.77 | 98.70 | 14.53 | 97.49 | 10.15 | 98.10 |
>
>
> Table 2: using far OOD prompt in fine-grained OOD task
> | Method | ID / OOD | CUB-100/ CUB-100 |  | Stanford-Cars-98 / Stanford-Cars-98 |  | Food-50 / Food-51 |  | Oxford-Pet-18 / Oxford-Pet-19 |  | Average |  |
> | :---: | :---: | :---: | :---: | :---: | :---: | :---: | :---: | :---: | :---: | :---: | :---: |
> |  |  | FPR95 $\downarrow$ | AUROC $\uparrow$ | FPR95 $\downarrow$ | AUROC $\uparrow$ | FPR95 $\downarrow$ | AUROC $\uparrow$ | FPR95 $\downarrow$ | AUROC $\uparrow$ | FPR95 $\downarrow$ | AUROC $\uparrow$ |
> | MCM |  | 83.58 | 67.51 | 83.99 | 68.71 | 43.48 | 91.75 | 63.92 | 84.88 | 68.72 | 78.21 |
> | Ours |  | 71.57 | 74.12 | 77.88 | 71.08 | 39.04 | 91.81 | 57.89 | 88.72 | 61.60 | 81.43 |
> | Ours(far prompt) |  | 78.27 | 72.24 | 75.20 | 73.22 | 38.23 | 91.95 | 69.00 | 88.37 | 65.18 | 81.45 |

---

> ### Author Response · Authors · 2023-11-18
> **Response to Reviewer WwTG (4/4)**
>
> >**Q5**. *Why use FPR95 instead of AUPR?*
> >
> >**Reply:** Firstly, in OOD detection community, FPR95 is more commonly used. Specifically, FPR95 is frequently utilized to evaluate the false positive rate at high true positive rates. Since DOS is capable of achieving a high true positive rate, using FPR95 is more appropriate in this context.
> >
> >Second, AUPR is often used to when evaluating unbalanced datasets (where the number of positive and negative samples varies greatly).  However, since we are following MCM's dataset setting, where the difference in the number of positive and negative samples is not substantial, using FPR95 is more suitable in this scenario.
> >
> >Thirdly, to investigate the performance of MCM and DOS in terms of AUPR, we report the results when positive samples are fewer than negative samples: ImageNet10 (ID) and ImageNet20 (OOD), with positive samples being half the number of negative ones. The specific results are as follows. Our DOS also outperforms MCM on the AUPR metric.
>
> Table: AUPR
> | Method | ID / OOD | ImageNet-10 / ImageNet-20 |  ImageNet-20 / ImageNet-10 |
> | :---: | :---: | :---: | :---: |
> | MCM |  | 97.62 | 99.02 |
> | Ours | |  98.37 | 99.07
>
> >---
>
> >**Q6**. *In Figure 7, why are far-ood and fine-grained-ood tested with 100s of outlier class labels but near-ood only with a maximum of 10?*
> >
> >**Reply:** We would like to clarify that the number of outlier class labels is 100 instead of 10 for the near OOD task. Specifically, we ask LLM to return 10 outlier class labels for **each ID class**, rather than giving 10 outlier class labels for **the entire dataset**. Thus the total of outlier classes is 100 for ImageNet10(ID).

---

> ### Author Response · Authors · 2023-11-20
> **Looking forward to your reply**
>
> Dear Reviewer WwTG,
>
> We sincerely appreciate your valuable feedback.
>
> As the deadline for the author-reviewer discussion phase is approaching, we would like to check if you have any other remaining concerns about our paper.
>
> We sincerely thank you for your dedication and effort in evaluating our submission. Please do not hesitate to let us know if you need any clarification or have additional suggestions.
>
> Best Regards,
>
> Authors.

---

> ### Author Response · Authors · 2023-11-22
> **Would you mind checking our responses and confirming whether you have any further questions?**
>
> Dear Reviewer WwTG,
>
> As the rebuttal discussion phase ends in less than 12 hours, we want to express our gratitude for your engagement thus far. We would like to kindly remind you that after the 22nd (AOE) we will not be able to answer any further questions you may have. We really want to check with you whether our response addresses your concerns during the author-reviewer discussion phase.
>
> Your feedback is really important to us. We eagerly await any potential updates to your ratings, as they play a critical role in the assessment of our paper. We sincerely hope our responses have addressed your concerns and provided satisfactory explanations. Your thoughtful evaluation greatly helps us improve the overall strength of our paper. We sincerely appreciate your dedication and time again.
>
> Best regards,
>
> Authors

---

### Official Review · Reviewer_vW8h · 2023-10-30

**Soundness:** 4 excellent
**Presentation:** 4 excellent
**Contribution:** 2 fair
**Rating:** 8
**Confidence:** 4

**Summary:**

The paper presents DOS, a method for improving zero-shot OOD detection for CLIP-like models. DOS can be summarized in two parts: firstly the generation of broad OOD labels using an existing LLM, and secondly the detection of OOD samples through a proposed DOS scoring function. Experimental results on a range of ID and OOD datasets show improvements over the baseline MCM and other commonly used OOD detection methods.

**Strengths:**

1. The paper is clearly written and the zero-shot OOD detection method can be easily, and widely, used in real-world applications of OOD detection.
2. The distinguishment of far, near, and fine-grain OOD label generation presents interesting and unique opportunities for future work.
3. Empirical results show impressive performance, even when compared against fine-tuned methods of OOD detection.

**Weaknesses:**

1. The reviewer would personally like to see additional experimental evaluations beyond CLIP models, such as ALIGN[1] or FLAVA[2].
2. Additional experiments with other standard OOD detection benchmarks such as CIFAR-10/CIFAR-100 (SVHN, LSUN, DTD, Places365) would give further empirical support for the methodology.

[1] Jia, Chao, Yinfei Yang, Ye Xia, Yi-Ting Chen, Zarana Parekh, Hieu Pham, Quoc Le, Yun-Hsuan Sung, Zhen Li, and Tom Duerig. "Scaling up visual and vision-language representation learning with noisy text supervision." In International conference on machine learning, pp. 4904-4916. PMLR, 2021.

[2] Singh, Amanpreet, Ronghang Hu, Vedanuj Goswami, Guillaume Couairon, Wojciech Galuba, Marcus Rohrbach, and Douwe Kiela. "Flava: A foundational language and vision alignment model." In Proceedings of the IEEE/CVF Conference on Computer Vision and Pattern Recognition, pp. 15638-15650. 2022.

**Questions:**

The reviewer would like some additional clarification regarding the T-SNE visualization in Section 4.4. In particular, it is unclear from initial viewing why the T-SNE visualization implies improved OOD detection performances, as one can similarly argue how the singlular clustering of OOD representations may lead to better OOD detection.

---

> ### Author Response · Authors · 2023-11-18
> **Response to Reviewer vW8h (1/2)**
>
> We thank the reviewer vW8h for the valuable feedback. We addressed all the comments. Please find the point-to-point responses below. Any further comments and discussions are welcomed!
>
> >**Q1**. *The reviewer would personally like to see additional experimental evaluations beyond CLIP models, such as ALIGN[1] or FLAVA[2].*
> >
> >**Reply:** Thank you for your valuable feedback! We believe that evaluations beyond CLIP models are crucial, and have therefore conducted experiments on ImageNet-1K using [the pretrained ALIGN model](https://huggingface.co/kakaobrain/align-base), with the results shown in the table below. By using ALIGN as the VLM, our DOS can also consistently improve the baseline MCM. In addition, when using ALIGN as the VLM, our DOS's performance is significantly better than MCM(an enhancement of 29.90% in FPR95). These results indicate that our DOS is more generalizable to different VLMs. We have added the above experiments in Section 4.2 and Appendix. B (Table. 5) in the reivsion.
>
>
> | Method | OOD Dataset |  |  |  |  |  |  |  | Average |  |
> | :---: | :---: | :---: | :---: | :---: | :---: | :---: | :---: | :---: | :---: | :---: |
> |  | iNaturalist |  | SUN |  | Places |  | Texture |  |  |  |
> |  | FPR95 $\downarrow$ |  AUROC $\uparrow$ | FPR95 $\downarrow$ | AUROC $\uparrow$ | FPR95 $\downarrow$ | AUROC $\uparrow$ | FPR95 $\downarrow$ | AUROC $\uparrow$ | FPR95 $\downarrow$ | AUROC $\uparrow$ |
> | MCM (ViT-B/16) | 30.91 | 94.61 | 37.59 | 92.57 | 44.69 | 89.77 | 57.77 | 86.11 | 42.74 | 90.77 |
> | Ours (ViT-B/16) | 7.10 | 98.32 | 16.72 | 96.49 | 26.48 | 94.00 | 69.57 | 80.65 | 29.97(+12.77) | 92.37(+1.60) |
> | MCM (ViT-L/14) | 28.38 | 94.95 | 29.00 | 94.14 | 35.42 | 92.00 | 59.88 | 84.88 | 38.17 | 91.49 |
> | Ours (ViT-L/14) | 6.54 | 98.45 | 14.52 | 97.10 | 22.28 | 94.94 | 67.89 | 81.51 | 27.81(+10.36) | 93.01(+1.52) |
> | MCM (ALIGN) | 60.63 | 89.39 | 53.20 | 89.17 | 61.05 | 85.58 | 59.63 | 83.94 | 58.63 | 87.02 |
> | DOS (ALIGN) | 9.54 | 98.07 | 15.38 | 96.77 | 25.24 | 93.89 | 64.75 | 81.57 | 28.73(+29.90) | 92.57(+5.55) |
>
> >---
>
> >**Q2**. *Additional experiments with other standard OOD detection benchmarks such as CIFAR-10/CIFAR-100 (SVHN, LSUN, DTD, Places365) would give further empirical support for the methodology.*
> >
> >**Reply:** Thanks for the constructive advice. We conducted experiments on the CIFAR-10/CIFAR-100 (SVHN, LSUN, DTD, Places365) datasets and the results are shown in the table below. Furthermore, we incorporated the widely used Energy and MaxLogit scores for comparison. All comparison methods utilize CLIP (ViT-B/16 as the visual encoder) as the VLM. Clearly, DOS achieves superior results on this standard OOD detection benchmark. We have added the above experiments in Section 4.2 and Appendix. C (Table. 6) in the reivsion.
>
> | ID Dataset | Method | OOD Dataset |  |  |  |  |  |  |  | Average |  |
> | :---: | :---: | :---: | :---: | :---: | :---: | :---: | :---: | :---: | :---: | :---: | :---: |
> |  |  | SVHN |  | LSUN |  | DTD |  | Places365 |  |  |  |
> |  |  | FPR95 $\downarrow$ | AUROC $\uparrow$ | FPR95 $\downarrow$ | AUROC $\uparrow$ | FPR95 $\downarrow$ | AUROC $\uparrow$ | FPR95 $\downarrow$ | AUROC $\uparrow$ | FPR95 $\downarrow$ | AUROC $\uparrow$ |
> | cifar10 | CLIPN | 53.28 | 74.20 | 27.89 | 92.72 | 3.58 | 98.93 | 9.82 | 96.98 | 23.64 | 90.63 |
> |  | Energy | 18.97 | 96.67 | 60.60 | 88.81 | 16.13 | 96.59 | 17.48 | 95.42 | 28.29 | 94.37 |
> |  | MaxLogit | 6.50 | 98.27 | 36.54 | 94.20 | 11.37 | 97.61 | 16.67 | 95.56 | 17.77 | 96.41 |
> |  | MCM | 3.98 | 99.03 | 5.12 | 98.72 | 16.35 | 96.44 | 36.55 | 90.79 | 15.50 | 96.25 |
> |  | Ours | 4.48 | 98.51 | 3.24 | 98.88 | 6.91 | 98.32 | 15.24 | 95.57 | **7.47** | **97.82** |
> | cifar100 | CLIPN | 71.72 | 68.20 | 84.42 | 80.90 | 37.74 | 90.92 | 51.06 | 87.25 | 61.24 | 81.82 |
> |  | Energy | 72.54 | 88.20 | 93.64 | 73.08 | 65.55 | 80.43 | 59.86 | 83.47 | 72.90 | 81.30 |
> |  | MaxLogit | 59.05 | 91.01 | 82.48 | 83.06 | 62.82 | 82.08 | 65.58 | 80.88 | 67.48 | 84.26 |
> |  | MCM | 64.45 | 89.96 | 47.26 | 91.69 | 90.30 | 73.61 | 98.42 | 61.37 | 75.11 | 79.16 |
> |  | Ours | 75.64 | 87.00 | 39.04 | 92.87 | 62.00 | 83.54 | 70.60 | 80.43 | **61.82** | **85.95** |
> | Average | CLIPN | 62.50 | 71.20 | 56.16 | 86.81 | 20.66 | 94.93 | 30.44 | 92.12 | 42.44 | 86.23 |
> |  | Energy | 45.76 | 92.44 | 77.12 | 80.95 | 40.84 | 88.51 | 38.67 | 89.45 | 50.60 | 87.84 |
> |  | MaxLogit | 32.78 | 94.64 | 59.51 | 88.63 | 37.10 | 89.85 | 41.13 | 88.22 | 42.63 | 90.34 |
> |  | MCM | 34.22 | 94.50 | 26.19 | 95.21 | 53.33 | 85.03 | 67.49 | 76.08 | 45.31 | 87.71 |
> |  | Ours | 40.06 | 92.76 | 21.14 | 95.88 | 34.46 | 90.93 | 42.92 | 88.00 | **34.65** | **91.88** |
>
> >---

---

> ### Author Response · Authors · 2023-11-18
> **Response to Reviewer vW8h (2/2)**
>
> >**Q3**. *The reviewer would like some additional clarification regarding the T-SNE visualization in Section 4.4. In particular, it is unclear from initial viewing why the T-SNE visualization implies improved OOD detection performances, as one can similarly argue how the singlular clustering of OOD representations may lead to better OOD detection.*
> >
> >**Reply:** From the T-SNE visualization, it's indeed challenging to discern the improved OOD detection performances of DOS. This is because the baseline MCM has already achieved an impressive 5% FPR95 on ImageNet10. The marginal 1% improvement is nuanced and not easily observable in the T-SNE visualization.
> >Nevertheless, the primary purpose of presenting the T-SNE visualization is not to highlight our advancements in OOD detection performance. Rather, our focus is to elucidate and demonstrate how the dreamed outlier class labels perform in OOD detection. We specifically show that samples that are more visually similar to the dreamed outlier classes are clustered together. This clustering result suggests that our DOS approach is more semantically interpretable in distinguishing outlier data.

---

> > ### Comment · Reviewer_vW8h · 2023-11-22
> > **Thanks to the authors for the responses**
> >
> > I would like to extend my thanks to the authors for addressing all the questions posed in my review and have updated my rating accordingly. However, I would like to encourage the author to rethink or reword Section 4.4, as the reviewer still finds the visualizations unintuitive for justifying how DOS is more explainable.

---

> > > ### Author Response · Authors · 2023-11-22
> > > **Thanks for your acknowledgement**
> > >
> > > Dear Reviewer vW8h,
> > >
> > > We are grateful for your positive feedback on our response and the increase of score.
> > >
> > > We also extend our sincere thanks for your additional suggestion. In accordance with your guidance, we have revised the visualization by 1) employing distinct colors to represent different OOD classes and 2) showing examples for the samples grouped into the same cluster, as well as the examples (generated by stable diffusion for visualization) for the dreamed OOD name that assigned with the corresponding cluster. The visualization shows that 1) OOD samples belonging to the same class tend to be clustered together and that 2) these clustered samples are assigned with the dreamed outlier class that is visually similar to them. We have updated the visualization and corresponding discussion in the main paper (see Sec 4.4 and Page 8).
> > >
> > > We hope that our response and revision could increase your confidence in our contribution and novelty. We would be immensely grateful for your support during the forthcoming discussion and decision-making process.
> > >
> > > Best Regards,
> > >
> > > Authors

---

> ### Author Response · Authors · 2023-11-20
> **Looking forward to your reply**
>
> Dear Reviewer vW8h,
>
> We sincerely appreciate your valuable feedback.
>
> As the deadline for the author-reviewer discussion phase is approaching, we would like to check if you have any other remaining concerns about our paper.
>
> We sincerely thank you for your dedication and effort in evaluating our submission. Please do not hesitate to let us know if you need any clarification or have additional suggestions.
>
> Best Regards,
>
> Authors.

---

### Official Review · Reviewer_yFD8 · 2023-11-02

**Soundness:** 2 fair
**Presentation:** 3 good
**Contribution:** 2 fair
**Rating:** 3
**Confidence:** 3

**Summary:**

This paper proposes a new approach to address the OOD detection problem. The author suggests that having knowledge of the categories of the OD instances can effectively improve the OOD detection performance. Furthermore, the author proposes using LLM to generate the category names for OD instances. Building upon this idea, the author designs a novel OOD detection algorithm. According to the experimental results, the method proposed in this paper demonstrates improved OOD detection performance.

**Strengths:**

1. The motivation of this paper is very innovative. In the rapidly developing field of LLM, introducing LLM into OOD detection tasks could indeed lead to significant improvements.

2. The author's writing is clear, explaining the starting point, specific methods, and experimental design of this article very clearly.

3. According to the author's experimental results, the proposed method in this article can indeed improve the effectiveness of OOD detection tasks.

**Weaknesses:**

1. Although it is a good idea to introduce LLM into the OOD detection task, the way it is introduced in this paper is somewhat rigid. The paper primarily utilizes LLM to generate names for OD samples, which are then employed for training purposes. However, there is a lack of effective measures to ensure the reasonability of the OD categories generated by LLM. This critical oversight significantly compromises the overall reliability and trustworthiness of the proposed method.

2. The analysis in this paper is insufficient. Firstly, considering the pivotal role played by LLM in the proposed method, it is crucial to explore the performance of various LLM models, rather than solely relying on a single model such as gpt-3.5. Conducting experiments with different LLM models could potentially yield diverse outcomes and provide a more comprehensive understanding of the approach's effectiveness. By limiting the analysis to just one model, the authors unintentionally overlook the possibility of alternative models delivering superior results.
Secondly, the practical implications of the categories generated by LLM for the ODs are not thoroughly examined. While these generated categories may possess semantic relevance, it is essential to assess the extent of overlap between the generated categories and the actual OD categories. Additionally, an in-depth analysis of the impact of these categories on the final accuracy of the OOD detection system is missing. Understanding the potential discrepancies and evaluating the influence of these categories on the system's overall performance is crucial for gauging the practical applicability and reliability of the proposed method.

**Questions:**

As shown in the weakness

---

> ### Author Response · Authors · 2023-11-18
> **Response to Reviewer yFD8 (1/2)**
>
> We thank the reviewer yFD8 for the valuable feedback. We addressed all the comments. Please find the point-to-point responses below. Any further comments and discussions are welcomed!
>
> >**Q1.1**. *Although it is a good idea to introduce LLM into the OOD detection task, the way it is introduced in this paper is somewhat rigid. The paper primarily utilizes LLM to generate names for OD samples, which are then employed for training purposes.*
> >
> >**Reply:** We would like to further explain the motivation and implementation of our method.
> >- **Motivation.** Our DOS is motivated by the observation that CLIP can achieve superior OOD detection performance when combined with ground truth OOD class labels. However, relying solely on the closed-set ID classes limits the original performance capabilities of CLIP. Since GT OOD class labels are unavailable in practice, we proposed employ the powerful knowledge of LLM to dream potential informative outlier class labels. This novel strategy enables us unleash extraordinary OOD detection capability in CLIP.
> >- **Implementation.** We would like to emphasize that our DOS **does not require any training**. All experiments are conducted in **zero-shot setting**, without training samples. This zero-shot setting has been clearly explained in both the abstract and the main text.
>
> >---
>
> >**Q1.2**. *However, there is a lack of effective measures to ensure the reasonability of the OD categories generated by LLM.*
> >
> >**Reply:** Our well-designed prompt requires that the OD categories returned by the LLM are visually resemble to the ID categories. Therefore, we have conducted **ablation experiments (see Sec.4.3 and Figure 6)** to ensure the reasonableness of OD categories that is `visually resemble` to ID categories. Specifically, different LLM prompts will lead the LLM to generate OD categories with diverse characteristics. We have designed two types of LLM prompts, one termed `visually irrelevant` and the other `visually dissimilar`, as well as maintaining the rest of the LLM prompts constant. Specifically, the `irrelevant` LLM prompt instructs the LLM to generate arbitrary outlier class labels for the ID class without adhering to a `visually resemble` constraint. Conversely, the `dissimilar` LLM prompt asks the LLM to dream outlier class labels for the ID class under a `visual dissimilar` constraint. The results on ImageNet10(ID) are shown in the table below.
>
> Table: ImageNet10(ID), ImageNet20(OOD)
> | Method | FPR95 $\downarrow$ | AUROC $\uparrow$ |
> | :---: | :---: | :---: |
> | MCM(Baseline) |  5.00 | 98.71 |
> | visually irrelevant |  5.73 | 98.63 |
> | visually dissimilar |  5.70 | 98.58 |
> | visually resemble(Ours) |  **4.00** | **99.09** |
>
> >Without the `visually resemble` constraint proposed in our DOS, the OOD performance degrades on both FPR95 and AUROC metrics, indicating the importance of the proposed constraint. This demonstrates the reasonability of the OD categories generated by the LLM under the constraint of `visual resemble`.
> >
> >We also provide a further analysis of our approach by investigating the overlap between the generated and ground-truth OOD names (as your suggestion). Please refer to Q2.2.
>
> >---
>
> >**Q2.1**. *Explore the performance of various LLM models.*
> >
> >**Reply:** Good suggestion. We conduct experiments with various LLMs to provide a more comprehensive understanding of the approach's effectiveness. Specifically, we use LLaMA2-7B or Claude2 to dream outlier class labels. The results on ImageNet10 (ID) are shown in the following table. All the variants, which use different LLMs, achieve better results than the baseline MCM. Moreover, Claude2 outperforms gpt-3.5-turbo in the FPR95 metric. These results demonstrate the universality of our method. We add the results in Fig. 6(c) in our revised paper.
>
> | Methods       | FPR95 $\downarrow$ | AUROC $\uparrow$ |
> | :-:         |         :-:       |         :-:         |
> | MCM   |        5.00%     |     98.71%     |
> | Ours(LLaMA2-7B)      |       4.1%      |     98.72%         |
> | Ours(Claude2)   |        3.53%     |     98.94%     |
> | Ours(gpt-3.5-turbo) |       4.00%      |     99.09%           |
>
> >---

---

> ### Author Response · Authors · 2023-11-18
> **Response to Reviewer yFD8 (2/2)**
>
> >**Q2.2**. *Assess the extent of overlap between the generated categories and the actual OD categories.*
> >
> >**Reply:** Good question. We found that our approach fewly hits the GT OOD classes. For example, the number of hitting classes is less than 3 when using ImageNet10 as the ID set and ImageNet20 as the OOD set.
> >
> >In fact, hitting the GT OOD class is impractical because the OOD data that models encounter can be diverse and unpredictable. This is why our method can be beneficial to the OOD community. Based on the visual features of ID classes, we ask LLM to dream potential outlier classes that could easily be confused with ID classes. Even without hitting the GT OOD, this approach can still enhance our performance in OOD detection.
> >
> >For example, suppose we have an ID class 'horse'. When the model encounters OOD data 'gazelle', it might mistakenly identify it as a 'horse'. If LLM returns the outlier class label 'deer', it is likely that the model will classify 'gazelle' as 'deer', thereby correctly identifying it as OOD data and consequently enhancing performance.
>
> >---
>
> >**Q2.3**. *Additionally, an in-depth analysis of the impact of these categories on the final accuracy of the OOD detection system is missing.*
> >
> >**Reply:** We have conducted experiments to evaluate the impact of generated categoties in two apsects.
> >- Ablation study of the `visually resemble` prompt: Firstly, varying constraints in prompts result in the generation of diverse OD categories. We thus conducted ablation experiments on `visually resemble`, please refer to Section4.3 in our paper. Essentially, this is ablation study about OD categories, emphasizing their influence on the final accuracy of the OOD detection system. We have demonstrated that OD categories visually similar to ID categories are beneficial for improving the accuracy of the OOD detection system, with results presented in the response to Q1.2.
> >- Ablation study of the number of OD categories: We conduct experiments to investigate the impact of the number of OD categories generated by LLM, i.e. $L$. For the far and fine-grained OOD detection, we instruct the LLM to return $L=100$, $L=300$, and $L=500$ OD categories for each ID dataset. For the near OOD detection, we ask the LLM to return OD categories in quantities of $1$, $3$, and $10$ for each ID class. Although $L$ influences DOS performance, DOS can consistently outperform the baseline MCM and achieve relatively stable performance when $L$ is not too small. The results are shown in the tables below.
>
> Table 1: Far OOD
> | # OD categories($L$) | FPR95 $\downarrow$ | AUROC $\uparrow$ |
> | :---: | :---: | :---: |
> | MCM(Baseline) |  2.70 | 99.40 |
> | Ours($L=100$) |  0.23 | 99.90 |
> | Ours($L=300$) |  0.22 | 99.90 |
> | Ours($L=500$) |  0.21 | 99.90 |
>
> Table 2: Near OOD, $N$ is the number of categories in the ID dataset.
> | # OD categories for each ID class | FPR95 $\downarrow$ | AUROC $\uparrow$ |
> | :---: | :---: | :---: |
> | MCM(Baseline) |  11.20 | 98.30 |
> | Ours($L=1\times N$) |  13.00 | 97.10 |
> | Ours($L=3\times N$) |  9.40 | 98.60 |
> | Ours($L=10\times N$) |  10.80 | 98.40 |
>
> Table 3: Fine-grained OOD
> | # OD categories($L$) | FPR95 $\downarrow$ | AUROC $\uparrow$ |
> | :---: | :---: | :---: |
> | MCM(Baseline) |  68.70 | 78.20 |
> | Ours($L=100$) |  62.20 | 81.00 |
> | Ours($L=300$) |  63.20 | 80.90 |
> | Ours($L=500$) |  61.60 | 81.40 |
>
>
> >Last, we present the names of OD categories generated by different prompts for a deeper understanding, taking ID clases 'horse' and 'swiss mountain dog' as an example. The OD categories given by different LLM prompts are as follows.
> >
> | LLM prompt | ID class: horse | ID class: swiss mountain dog |
> | :---: | :---: | :---: |
> | visually dissimilar |  pineapple, laptop, mountain | sunflower, laptop, beach umbrellas |
> | visually irrelevant |  pineapple, laptop, sunglasses | tulip flower, laptop computer, beach ball |
> | Ours(visually resemble) |  zebra, giraffe, deer | brown bear, alpine landscape, hiking boots |
>
> >From the above OD categories and experimental results, we can draw the following conclusions: With visual resemblance constraints, LLM will return OD categories that are visually similar to the ID classes, which can help improve the accuracy of the OOD detection system. On the contrary, the generated OD categories are not beneficial for OOD detection without visual resemblance constraints.

---

> ### Author Response · Authors · 2023-11-20
> **Looking forward to your reply**
>
> Dear Reviewer yFD8,
>
> We sincerely appreciate your valuable feedback.
>
> As the deadline for the author-reviewer discussion phase is approaching, we would like to check if you have any other remaining concerns about our paper.
>
> We sincerely thank you for your dedication and effort in evaluating our submission. Please do not hesitate to let us know if you need any clarification or have additional suggestions.
>
> Best Regards,
>
> Authors.

---

> ### Author Response · Authors · 2023-11-22
> **Would you mind checking our responses and confirming whether you have any further questions?**
>
> Dear Reviewer yFD8,
>
> As the rebuttal discussion phase ends in less than 12 hours, we want to express our gratitude for your engagement thus far. We would like to kindly remind you that after the 22nd (AOE) we will not be able to answer any further questions you may have. We really want to check with you whether our response addresses your concerns during the author-reviewer discussion phase.
>
> Your feedback is really important to us. We eagerly await any potential updates to your ratings, as they play a critical role in the assessment of our paper. We sincerely hope our responses have addressed your concerns and provided satisfactory explanations. Your thoughtful evaluation greatly helps us improve the overall strength of our paper. We sincerely appreciate your dedication and time again.
>
> Best regards,
>
> Authors

---

### Official Review · Reviewer_5Vrp · 2023-11-10

**Soundness:** 4 excellent
**Presentation:** 3 good
**Contribution:** 3 good
**Rating:** 6
**Confidence:** 4

**Summary:**

This paper tackles the problem of zero-shot OOD detection. Following the previous CLIP-based OOD detection methods (MCM), this paper finds that adding OOD label space with ID classes could boost performance. To this end, the authors propose to leverage LLM to generate prompts to dream about OOD classes. Further, a new scoring function is proposed based on the proportionality between potential outlier and ID class labels. Experiments show the performance gains compared to MCM.

**Strengths:**

- Using LLMs to generate prompts for OOD classes is interesting and it is based on an empirical study that using OOD classes w/ ID classes will improve the performance.
- The proposed method is zero-shot, training-free;
- Ablation study on score function; Number of OOD classes has been conducted and explained.

**Weaknesses:**

Overall, I think the paper is interesting to the community for discussion. Yet, I still have some questions or concerns that want the authors to address in rebuttal.
1. For Figure 1, I get the high-level idea that adding OOD classes with/ ID classes helps the performance boost. Can you add those GT OOD classes w/ ID classes in your Table 1,2,3,4 for the Oracle experiment? It can help better understand the upper bound of your approaches.
2. For the scoring function, I don't see much motivation or justification for this scoring function design. Also, in Figure 6(a), the performance gains from S_DOS to S_MSP are minor. Can you elaborate more on this scoring function design?
- Also, it will be interesting to test your method on other well-known scoring functions such as Max logit score; Energy function; gradients, etc.
3. In table 2, the performance of the Texture dataset is not good. Do you have any insight on the reason?
4. For your generated OOD class prompts, can you conduct some similarity measures between your prompts and GTs? I would like to see how much chance LLMs can hit the GT OOD classes.

**Questions:**

Please refer to the weakness

---

> ### Author Response · Authors · 2023-11-18
> **Response to Reviewer 5Vrp (1/3)**
>
> We thank the reviewer 5Vrp for the valuable feedback. We addressed all the comments. Please find the point-to-point responses below. Any further comments and discussions are welcomed!
>
> >**Q1**. *Can you add those GT OOD classes w/ ID classes in your Table 1,2,3,4 for the Oracle experiment?*
> >
> >**Reply:** Thanks for the constructive advice. We have added the results with the GT OOD classes in Tables 1-4. Note that several OOD datasets used in Tables 1 and 2 are subsets of the entire datasets without providing GT: including iNaturalist, SUN, and Places, curated by [[1]](https://github.com/deeplearning-wisc/large_scale_ood#out-of-distribution-dataset). Thus, the GT names for them cannot be obtained. If we use the GT of the entire  datasets, there will be an overlap with the ID dataset. We have solved this problem for the Places dataset by extracting the GT names from the image file names. However, this solution is not available for the iNaturalist and SUN datasets. We thus did not report the GT results for the iNaturalist and SUN datasets in Table 1 and 2.
> >
> >As shown in Table 1-4, after adding GT OOD classes, the performance is superior, verifying the motivation of this paper. In some scenarios, our DOS can achieve similar results to the one using GT, as shown in Table 1. We have included these results in the revised paper.
>
> [1] Huang et al. MOS: Towards Scaling Out-of-distribution Detection for Large Semantic Space. In CVPR, 2021.
>
> Table 1: We report the average values of metrics after adding the GT OOD classes to four ID datasets. ID datasets: CUB-200-2011, STANFORD-CARS, Food-101 and Oxford-IIIT Pet. For detailed results of each ID dataset, please refer to Table 1 in our paper.
> | ID Dataset | Method | OOD Dataset |   |  |  |
> | :---: |  :---: | :---: | :---: | :---: | :---: |
> |  |  | Places |  | Texture |  |  |  |
> |  |  | FPR95 $\downarrow$ | AUROC $\uparrow$ | FPR95 $\downarrow$ | AUROC $\uparrow$ | FPR95 $\downarrow$ | AUROC $\uparrow$ |
> | Average | MCM | 2.72 | 99.40 | 2.96 | 99.29 |
> |  | Ours |  **0.17** | **99.96** | **0.64** | **99.75** |
> |  | GT |  0.12 | 99.97 | 0.17 | 99.95 |
>
>
> Table 2: ID dataset: ImageNet-1K
> | Method | OOD Dataset |   |  |  |
> |  :---: | :---: | :---: | :---: | :---: |
> |   | Places |  | Texture |
> |   |FPR95 $\downarrow$ | AUROC $\uparrow$ | FPR95 $\downarrow$ | AUROC $\uparrow$ |
> | MCM (ViT-B/16) | 44.69 | 89.77 | **57.77** | **86.11** |
> | Ours (ViT-B/16) |  **26.48** | **94.00** | 69.57 | 80.65 |
> | GT (ViT-B/16) |  13.24 | 96.96 | 24.29 | 95.04 |
>
> Table 3: Near OOD
> | Method | ID / OOD | ImageNet-10 / ImageNet-20 |  | ImageNet-20 / ImageNet-10 |  | Average |  |
> | :---: | :---: | :---: | :---: | :---: | :---: | :---: | :---: |
> |  |  | FPR95 $\downarrow$ | AUROC $\uparrow$ | FPR95 $\downarrow$ |  AUROC $\uparrow$ | FPR95 $\downarrow$ |  AUROC $\uparrow$ |
> | MCM |  | 5.00 | 98.71 | 17.40 | 97.87 | 11.20 | 98.29 |
> | Ours |  | **4.00** | **99.09** | **14.80** | **98.01** | **9.40** | **98.55** |
> | GT |  | 0.20 | 99.80 | 0.20 | 99.93 | 0.20 | 99.87 |
>
> Table 4: Fine-grained OOD
> | Method | ID / OOD | CUB-100 / CUB-100 |  | Stanford-Cars-98 / Stanford-Cars-98 |  | Food-50 / Food-51 |  | Oxford-Pet-18 / Oxford-Pet-19 |  | Average |  |
> | :---: | :---: | :---: | :---: | :---: | :---: | :---: | :---: | :---: | :---: | :---: | :---: |
> |  |  | FPR95 $\downarrow$ | AUROC $\uparrow$ | FPR95 $\downarrow$ | AUROC $\uparrow$ | FPR95 $\downarrow$ | AUROC $\uparrow$ | FPR95 $\downarrow$ | AUROC $\uparrow$ | FPR95 $\downarrow$ | AUROC $\uparrow$ |
> | MCM |  | 83.58 | 67.51 | 83.99 | 68.71 | 43.48 | 91.75 | 63.92 | 84.88 | 68.72 | 78.21 |
> | Ours |  | **71.57** | **74.12** | **77.88** | **71.08** | **39.04** | **91.81** | **57.89** | **88.72** | **61.60** | **81.43** |
> | GT |  | 61.23 | 81.42 | 58.31 | 83.71 | 11.34 | 97.79 | 29.17 | 95.58 | 40.01 | 89.63 |
>
> >---

---

> ### Author Response · Authors · 2023-11-18
> **Response to Reviewer 5Vrp (2/3)**
>
> >**Q2.1**. *Can you elaborate more on this scoring function design?*
> >
> >**Reply:** We elaborate on the design of our DOS score from two aspects:
> >- **How to utilize dreamed outlier class labels:** The utilization of the dreamed class is a crucial aspect. Based on the MSP score, an intuitive idea is to incorporate the dreamed class into the denominator（i.e., $S_{\text{MSP}}(x) = \max_{i \in [1, K]} \frac{e^{s_i(x)/\tau}}{\sum_{j=1}^{K+L} e^{s_j(x)/\tau}}$). However, in this case, the dreamed class only functions in the denominator, which doesn't significantly impact the final score distribution, implying that the dreamed class is not fully utilized. To amplify the role of the dream class, DOS further subtracts the second item($- \max_{k \in [K+1, K+L]} \frac{\beta e^{s_k(x)/\tau}}{\sum_{j=1}^{K+L} e^{s_j(x)/\tau}}$). It stems from an intuition: samples visually similar to the dreamed class should have lower scores, thus making it easier to distinguish between the ID and OOD score distribution.
> >- **How to balance the number of dreamed outlier class labels:** When L is large, the dreamed outlier class, merely being in the denominator, already significantly influences the overall score distribution. If we set $\beta=1$, this may lead to the dreamed outlier class overly influencing the score distribution, potentially causing a decline in performance. Therefore, it is reasonable to adaptively adjust $\beta$ based on the size of L. We have thus designed $\beta=\frac{K}{K+L}$, ensuring a balanced influence of the dreamed outlier class.
> >
> >In summary, the design of $S_\text{DOS}$ jointly considers two key aspects: the utilization of dreamed outlier class labels and the balancing of their quantity. In addition, expriments show that our $S_\text{DOS}$ is universal and **does not require hyperparameter adjustments** based on different ID/OOD datasets. It can achieve excellent experimental results across various datasets. We have included the above discussion in Appendix.A in the reivsion.
>
> >---
>
> >**Q2.2**. *In Figure 6(a), the performance gains from S_DOS to S_MSP are minor.*
> >
> >**Reply:** We would like to clarify that it is non-trival to obtain improvement on the ImageNet10/20 setting. Specifically, the baseline MCM has attained an impressive 5% FPR95 on ImageNet10, and it is not easy to improve further. Therefore, the performance gained from $S_\text{DOS}$ (4%) is not minor compared to $S_\text{MSP}$ (6.5%).
>
> >---
>
> >**Q2.3**. *Also, it will be interesting to test your method on other well-known scoring functions such as Max logit score; Energy function; gradients, etc.*
> >
> >**Reply:** Good suggestion. We further conducted experiments with Max logit and Energy score. Specifically, based on dreamed outlier class labels, we design $S_\text{MaxLogit}$ and $S_\text{Energy}$ as follows:
> \begin{equation}
>     S_{\text{Energy}}(x) = -T \left(\log \sum_{i=1}^K e^{s_i(x) / T} - \log \sum_{j=K+1}^L e^{s_j(x) / T}\right)
> \end{equation}
> >
> >\begin{equation}
>     S_{\text{MaxLogit}}(x) = \max_{i \in [1, K]} s_i(x)- \max_{j \in [K+1, K+L]} s_j(x),
> \end{equation}
> >
> >We test $S_{\text{Energy}}$ with varying choices of T(0.01, 1, 10, 100, 1000) and the best reported result is obtained with T=0.01. The comparison of different score function on ImageNet10(ID) are reported the table below. Our DOS outperforms $S_{\text{Energy}}$ and $S_{\text{MaxLogit}}$.  We have added the above experiments in Section 4.3 (Fig. 6(a)) in the reivsion.
>
> Table: ImageNet10 (ID), ImageNet20 (OOD)
> | Score     | FPR95 $\downarrow$ | AUROC $\uparrow$     |
> | :-:         |         :-:       |         :-:         |
> | $S_\text{MSP}$      |       6.50%      |     98.62%         |
> | $S_\text{Energy}$   |       28.90%     |     95.60%     |
> | $S_\text{MaxLogit}$ |       31.90%      |     94.88%           |
> | MCM       |       5.00%      |       98.71%         |
> | $S_\text{DOS}$       |       **4.00%**      |     **99.09%**         |
>
> >---

---

> ### Author Response · Authors · 2023-11-18
> **Response to Reviewer 5Vrp (3/3)**
>
> >**Q3**. *In table 2, the performance of the Texture dataset is not good. Do you have any insight on the reason?*
> >
> >**Reply:** Good question. Texture is a unique dataset consisting entirely of textural images. Since our designed prompt is generic and not optimized for specific OOD datasets (e.g., asking LLM to return visually textural categories), it results in the dreamed outlier class labels being visually irrelevant to texture. Consequently, the performance on the Texture dataset is not satisfactory.
>
>
> >**Q4**. *For your generated OOD class prompts, can you conduct some similarity measures between your prompts and GTs? I would like to see how much chance LLMs can hit the GT OOD classes.*
> >
> >**Reply:** Good question. We found that our approach fewly hits the GT OOD classes. For example, the number of hitting classes is less than 3 when using ImageNet10 as the ID set and ImageNet20 as the OOD set.
> >
> >In fact, hitting the GT OOD class is impractical because the OOD data that models encounter can be diverse and unpredictable. This is why our method can be beneficial to the OOD community. Based on the visual features of ID classes, we ask LLM to dream potential outlier classes that could easily be confused with ID classes. Even without hitting the GT OOD, this approach can still enhance our performance in OOD detection.
> >
> >For example, suppose we have an ID class 'horse'. When the model encounters OOD data 'gazelle', it might mistakenly identify it as a 'horse'. If LLM returns the outlier class label 'deer', it is likely that the model will classify 'gazelle' as 'deer', thereby correctly identifying it as OOD data and consequently enhancing performance.

---

> ### Author Response · Authors · 2023-11-20
> **Looking forward to your reply**
>
> Dear Reviewer 5Vrp,
>
> We sincerely appreciate your valuable feedback.
>
> As the deadline for the author-reviewer discussion phase is approaching, we would like to check if you have any other remaining concerns about our paper.
>
> We sincerely thank you for your dedication and effort in evaluating our submission. Please do not hesitate to let us know if you need any clarification or have additional suggestions.
>
> Best Regards,
>
> Authors.

---

> ### Author Response · Authors · 2023-11-22
> **Would you mind checking our responses and confirming whether you have any further questions?**
>
> Dear Reviewer 5Vrp,
>
> As the rebuttal discussion phase ends in less than 12 hours, we want to express our gratitude for your engagement thus far. We would like to kindly remind you that after the 22nd (AOE) we will not be able to answer any further questions you may have. We really want to check with you whether our response addresses your concerns during the author-reviewer discussion phase.
>
> Your feedback is really important to us. We eagerly await any potential updates to your ratings, as they play a critical role in the assessment of our paper. We sincerely hope our responses have addressed your concerns and provided satisfactory explanations. Your thoughtful evaluation greatly helps us improve the overall strength of our paper. We sincerely appreciate your dedication and time again.
>
> Best regards,
>
> Authors

---

### Author Response · Authors · 2023-11-18
**A General Response to All Reviewers**

**We would like to thank all the reviewers for their thoughtful suggestions on our paper.**

**We are glad that the reviewers generally have positive impressions of our work**, including **(1)** interesting paper and innovative motivation(5Vrp, yFD8, vW8h); **(2)** Simple method and easy to understand(5Vrp, WwTG); **(3)** comprehensive experiments and sufficient ablation study (5Vrp, WwTG); **(4)** impressive performance (yFD8, vW8h);  **(5)**  clearly written (yFD8, vW8h, WwTG);

**In the rebuttal period, we have provided detailed responses to all the comments and questions point-by-point.** We also have revised the paper accordinglly, which help us significantly improve the overall quality. The main modifications can be summarized as:
 - Add GT OOD classes w/ ID classes in Table 1,2,3,4 (5Vrp);
 - Add other well-known scoring functions in Fig. 6(a) (5Vrp);
 - Elaborate more on $S_\text{DOS}$ design in Appendix. A.
 - Add new empirical evaluations with various LLMs in Fig. 6(c) (yFD8);
 - Add new empirical evaluations beyond CLIP in Appendix. B (vW8h);
 - Add other standard OOD benchmarks in Appendix. C (vW8h);
 - Revise visualization for more explainable in Fig. 8 (vW8h);
 - Add more baselines and make our experiments be consistent in Table 1,2,3,4 (WwTG);
 - Further clarify ID /OOD datasets setting in Sec4.1 (WwTG)

Lastly, we would appreciate all reviewers’ time again. Would you mind checking our response and confirming whether you have any further questions? **We are anticipating your post-rebuttal feedback!**

---

### Meta-Review · Area_Chair_FxCc · 2023-12-18

**Metareview:**

This paper builds on methods utilizing CLIP-style backbones for out-of-distribution (OOD) detection, and specifically noting that when given OOD label-space samples the performance tends to improve. Since such labels may not be available a-priori, the authors propose to use a large language model (LLM) to generate such samples, specifically designing the prompt to elicit such labels and also proposing a slight modification of the score function. Results are shown across a range of datasets with far, near, and fine-grained OOD settings.

 Overall, while the reviewers found the idea somewhat interesting in the context of the current LLM excitement, there were a range of weaknesses brought up including 1) significance, motivation, and justification for the scoring function (5Vrp), 2) Lack of uniform improvement, e.g. in the case of the texture dataset (5Vrp), 3) Lack of analysis of similarity between the prompt outputs and ground truth OOD classes (5Vrp, yFD8), 4) Reliance on a single model (CLIP) and LLM (yFD8), and 5) Lack of generality of the method, requiring per-setting prompts with reduced performance when this is not the case (WwTG). Overall the scores were very mixed as a result (6,3,8,3).

  The authors provided a rebuttal, which did address some portions of the weaknesses, including additional experiments with another backbone and other LLMs. However, a range of weaknesses still remain, and in fact some are shown to be valid such as the reliance of per-setting prompts, where utilizing a single prompt significantly reduces performance and in several cases makes it perform worse than MCM. The fact that the method also does not work in some domains (textures) also raises concern that the method is not general enough. Considering the paper, reviews, rebuttal, and discussion, this paper has not reached the level of acceptance. While the idea is interesting, it essentially leverages the knowledge within LLMs in a way that is simple and sensitive to the prompt. Further, the scoring mechanism introduced is a fairly simple adaptation designed for this framework, rather than a general method. As a result, the overall contributions are not sufficient, and we recommend that the authors significantly improve these aspects for future submission.

**Justification For Why Not Higher Score:**

Overall, the idea is a simple empirical observation that LLMs can perform a small sub-task important to OOD detection (identification of OOD samples) and the execution and contributions leave something to be desired (e.g. the score change highlighted is relatively specific to the framework proposed and not a general improvement). Crucially, the method is specific to the prompt, which is designed per OOD setting, and therefore the generality of the method is not strong or well-analyzed.

**Justification For Why Not Lower Score:**

N/A

---

### Decision · Program_Chairs · 2024-01-16

Reject